# Temporal Dynamics Aware Adversarial Attacks on Discrete-Time Graph Models

## Abstract

Real-world graphs such as social networks, communication networks, and rating networks are constantly evolving over time. Many architectures have been developed to learn effective node representations using both graph structure and its dynamics. While the robustness of static graph models is well-studied, the vulnerability of the dynamic graph models to adversarial attacks is underexplored. In this work, we design a novel adversarial attack on discrete-time dynamic graph models where we desire to perturb the input graph sequence in a manner that preserves the temporal dynamics of the graph. To this end, we motivate a novel Temporal Dynamics-Aware Perturbation (TDAP) constraint, which ensures that perturbations introduced at each time step are restricted to only a small fraction of the number of changes in the graph since the previous time step. We present a theoretically-grounded Projected Gradient Descent approach for dynamic graphs to find the effective perturbations under the TDAP constraint. Experiments on two tasks — dynamic link prediction and node classification, show that our approach is up to 4x more effective than the baseline methods for attacking these models. We also consider the practical online setting where graph snapshots become available in real-time and extend our attack approach to use Online Gradient Descent for performing attacks under the TDAP constraint. In this more challenging setting, we demonstrate that our method achieves up to 5x superior performance when compared to representative baselines.

## 1 Introduction

Graph Neural Networks (GNNs) have been shown to be vulnerable to adversarial perturbations (Jin et al., 2020; Bojchevski & Günnemann, 2019; Dai et al., 2018; Wu et al., 2019; Zügner et al., 2018; Ma et al., 2020a). This has raised major concerns against their use in important industrial applications such as friend/product recommendation (Ying et al., 2018; Sankar et al., 2021; Tang et al., 2020) and fraud detection (Zhao et al., 2021; Hooi et al., 2017). However, these advancements in designing attack and defense mechanisms have predominantly focused on GNN models for static, non-evolving graphs. In reality, the graph structure evolves with time as new interactions happen and new connections are formed (Leskovec et al., 2007; Kossinets & Watts, 2006). GNN models that incorporate the temporal information are shown to outperform their static counterparts in modeling dynamic networks on tasks such as predicting link existence in the future (Kazemi et al., 2020; Pareja et al., 2020; Sankar et al., 2020; Goyal et al., 2018; Chen et al., 2018).

However, the vulnerability of dynamic graph models to adversarial perturbations is less studied. The design of adversarial attacks for dynamic graphs is challenging for two reasons — (1) Attacks must simultaneously optimize both the edge(s) to perturb and when to perturb them, and more importantly, (2) Attacks must preserve the original graph evolution after perturbation in order to be less detectable. Attacks that disturb original graph evolution are not desired since they can be detected as anomalies by defense mechanisms, e.g. graph anomaly detection methods (Akoglu et al., 2015; Bunke et al., 2007; Cai et al., 2021). Therefore, it is crucial to formulate adversarial attacks over snapshots such that they do not significantly alter the original change in the graph structure.

In this work, we introduce a novel **T**emporal **D**ynamics-**A**ware **P**erturbation (TDAP) constraint to formulate evolution-preserving attacks on discrete-time dynamic graphs. This constraint asserts that the number of modifications added at the current timestep should only be a small fraction of

| Method | Dynamic | White-box | Evasion | Targeted | TDAP | Online |
|---|---|---|---|---|---|---|
| PGD (Xu et al., 2019b) | | ✓ | ✓ | ✓ | | |
| IG-JSMA (Xu et al., 2019b) | | ✓ | ✓ | ✓ | | |
| Fan et al. Fan et al. (2020) | ✓ | | ✓ | | | |
| Dyn-Backdoor (Chen et al., 2021a) | ✓ | ✓ | | | | |
| TGA (Chen et al., 2021b) | ✓ | ✓ | ✓ | ✓ | | |
| TD-PGD (proposed) | ✓ | ✓ | ✓ | ✓ | ✓ | ✓ |

Table 1: Comparison of our attack with existing works on graph adversarial attacks. Note that an attack is TDAP if the perturbations made are aware of the temporal dynamics.

the actual number of changes with respect to the preceding timestep. We show theoretically that perturbations made under TDAP constraint preserves the rate of change both in the structural and the embedding spaces. To find effective attacks under this proposed constraint, we consider a targeted, white-box, and evasion setting. As noted in Table 1, no prior works exist that can find attacks under our novel setting. Thus, we present a theoretically-grounded Temporal Dynamics-aware Projected Gradient Descent (TD-PGD) approach. The locality of the constraint in time allows us to easily extend this approach to find attacks in a more practical online setting (Mladenovic et al., 2021) that has not been studied before for dynamic graphs. Here, perturbations are found in real-time without any knowledge of the future snapshots. Our contributions can be summarized as follows:

1. We introduce a novel Temporal Dynamics-Aware Perturbation (TDAP) constraint to make perturbations in discrete-time dynamic graphs that preserves the evolution of the graphs.
2. We present a theoretically-grounded PGD-based white-box attack to find effective attacks on dynamic graphs under the novel TDAP constraint in both offline and online settings.
3. We show that TD-PGD outperforms the baselines across 4 different datasets and 3 victim models on both dynamic link prediction and node classification tasks.
4. We test the attacks on dynamic graphs in a novel online setting and show that the online version of TD-PGD shows improved performance over existing baselines.

## 2 RELATED WORK

**Representation Learning for Dynamic Graphs.** GNNs have been combined with sequential modeling architectures (Kazemi et al., 2020) to model dynamic graphs. For instance, discrete-time graphs have been modeled by using GNNs and RNNs together in a pipeline (Narayan & Roe, 2018; Manessi et al., 2020) or an embedded manner (Chen et al., 2018; Pareja et al., 2020). Attention-based models have also been proposed to jointly encode the graph structure and its dynamics (Sankar et al., 2020). For continuous-time graphs, both RNN (Kumar et al., 2019; Trivedi et al., 2017; 2019; Ma et al., 2020b) and attention-based models (Rossi et al., 2020; Xu et al., 2020) have been proposed such that embeddings are updated in real time upon an occurrence of a new event.

**Adversarial attacks on graphs.** Static GNNs are known to be vulnerable to adversarial attacks in different settings (Jin et al., 2020). White-box attacks are studied assuming complete knowledge of the underlying model Wu et al. (2019); Xu et al. (2019b). Limiting the model knowledge, gray-box (Zügner et al., 2018) and black-box attacks (Dai et al., 2018) have also been proposed. In comparison, the literature on adversarial attacks for dynamic graphs is scarce. Time-aware Gradient Attack (TGA) (Chen et al., 2021b) is a white-box evasion attack that greedily selects the perturbations across time under a budget constraint. In addition, attacks to poison training data (Chen et al., 2021a) and black-box attacks using RL approaches (Fan et al., 2020) have also been proposed.

**Imperceptible perturbations.** The most common strategy to formulate imperceptible attacks on graphs is to bound the total number of perturbations. In the case of dynamic graphs, perturbations must preserve the temporal flow to be imperceptible. Traditional anomaly detection algorithms flag an instance to be anomalous if distance between consecutive snapshots crosses a threshold (Akoglu et al., 2015). In particular, Graph Edit Distance and Hamming distance between adjacency matrices have been used to monitor communication networks (Shoubridge et al., 2002; Bunke et al., 2007). Neural approaches have looked at the consecutive change in the embedding space to detect anomalies without feature extraction (Goyal et al., 2018; Cai et al., 2021).

## 3 METHODOLOGY

**Problem** Let $\mathcal{G}_1, \mathcal{G}_2, \cdots, \mathcal{G}_T$ be the original graph snapshots and $\mathcal{G}'_1, \mathcal{G}'_2, \cdots, \mathcal{G}'_T$ be the corresponding perturbed snapshots. Note that $\mathcal{G}_i = (\boldsymbol{\mathcal{X}}_i, \boldsymbol{\mathcal{A}}_i)$ where $\boldsymbol{\mathcal{X}}_i, \boldsymbol{\mathcal{A}}_i$ are the node features and the adjacency matrix for snapshot $i$, respectively. Also, let $\mathcal{M}$ be a victim dynamic graph model that we want to attack and let $f_{\mathcal{M}}$ be a function that generates the corresponding node embeddings of $\mathcal{G}_t$ given $\mathcal{G}_{1:t-1}$. Let $y_{\text{task}}$ be the actual labels for a given task (for dynamic link prediction, these correspond to binary labels representing link existence in the future snapshot).

Then, the objective of the attacker is to introduce structural perturbations $\mathbf{S}_t = \boldsymbol{\mathcal{A}}'_t - \boldsymbol{\mathcal{A}}_t$ at each timestep $t < T$ such that the model inference at timestep $T$ for the target entities $E_{tg}$ deteriorates. More formally, the attacker solves the following optimization problem:

$$\max_{\boldsymbol{\mathcal{A}}'_1, \boldsymbol{\mathcal{A}}'_2, \cdots, \boldsymbol{\mathcal{A}}'_{T-1}} \mathcal{L}_{\text{task}}\left(\widehat{y}_{\text{task}}(f_{\mathcal{M}}(\boldsymbol{\mathcal{A}}'_{1:T-1})), y_{\text{task}}, E_{tg}\right) \tag{1}$$
$$\text{such that} \quad \mathcal{C}(\boldsymbol{\mathcal{A}}'_{1:T-1}) \text{ holds}$$

for some constraint function $\mathcal{C}$ on the perturbed adjacency matrices $\boldsymbol{\mathcal{A}}'_t$ for each time $t$. Here, $\widehat{y}_{\text{task}}$ denotes the predicted labels for the given task and $\mathcal{L}_{\text{task}}$ is a task-specific loss, for example, a binary cross entropy (CE) loss for link prediction.

The constraint function $\mathcal{C}$ is designed to ensure imperceptibility of the adversarial perturbations. In the literature, a budget constraint has been widely used to enforce imperceptibility in graphs (Dai et al., 2018) and computer vision (Goodfellow et al., 2014). However, this constraint only bounds the total amount of perturbations that can be introduced by an attacker. When the input is dynamic, as in the case of dynamic graphs, the perturbations should be constrained in the context of how the input evolves. However, since the budget constraint completely ignores the graph dynamics, it could lead to a drastic change in the evolution trend of the graph and thus, making the attacks easily detectable. For instance, with the budget constraint, all the perturbations can be made at a single time step, leading to an anomalous spike, which would be easily detected as a possible attack by graph anomaly detection methods for dynamic graphs (Akoglu et al., 2015; Shoubridge et al., 2002; Bunke et al., 2007). Thus, a constraint is desired that can ensure that the introduced perturbations do not disrupt the evolving trend of the dynamic graphs.

### 3.1 TEMPORAL DYNAMICS-AWARE PERTURBATION (TDAP) CONSTRAINT

The simplest measure to study evolution is to consider the change in the input between consecutive time steps. Thus, for a discrete-time input $\{\mathbf{x}_t\}$, this corresponds to considering the discrete-time differential norm at time $t$, given by $d\mathbf{x}_t = \|\mathbf{x}_t - \mathbf{x}_{t-1}\|$. Then, we propose

**Proposition 1** *The number of perturbations introduced to input $\mathbf{x}$ at time step $t$ must not be more than a fraction $\epsilon$ times the differential at $t$, i.e. $TDAP(\epsilon) := \|\mathbf{x}'_t - \mathbf{x}_t\| \le \epsilon d\mathbf{x}_t \ \forall t$.*

For the case of dynamic graphs, when the graph structure evolves (for example, in social networks and transaction networks (Pareja et al., 2020; Sankar et al., 2020)), this constraint becomes $\|\boldsymbol{\mathcal{A}}'_t - \boldsymbol{\mathcal{A}}_t\|_1 \le \epsilon d\boldsymbol{\mathcal{A}}_t$. Alternatively, a dynamic graph may also involve a temporally-evolving signal at each node (Rozemberczki et al., 2021; Li et al., 2017; Panagopoulos et al., 2021), in which case, this constraint becomes $\|\boldsymbol{\mathcal{X}}'_t - \boldsymbol{\mathcal{X}}_t\|_1 \le \epsilon d\boldsymbol{\mathcal{X}}_t$.

In this work, we focus on dynamic graph structures such that the constraint $\mathcal{C}$ (in Equation 1) for the perturbations is a TDAP constraint. The optimization problem for the attacker, thus, becomes

$$\max_{\boldsymbol{\mathcal{A}}'_1, \boldsymbol{\mathcal{A}}'_2, \cdots, \boldsymbol{\mathcal{A}}'_{T-1}} \mathcal{L}_{\text{task}}\left(\widehat{y}_{\text{task}}(f_{\mathcal{M}}(\boldsymbol{\mathcal{A}}'_{1:T-1})), y_{\text{task}}, E_{tg}\right) \tag{2}$$
$$\text{such that } \forall t \in (1, T): \frac{\|\boldsymbol{\mathcal{A}}'_t - \boldsymbol{\mathcal{A}}_t\|}{\|\boldsymbol{\mathcal{A}}_t - \boldsymbol{\mathcal{A}}_{t-1}\|} \le \epsilon$$
$$\|\boldsymbol{\mathcal{A}}'_1 - \boldsymbol{\mathcal{A}}_1\| \le \varepsilon_1,$$

where $\epsilon, \varepsilon_1$ are given parameters for this optimization. We use $\|\cdot\|$ to denote the 1-norm of the matrix flattened into a vector, unless otherwise mentioned.

**Implications.** We show that TDAP constraint has the following implications on the perturbations:

1. **Perturbations under TDAP constraint preserves the average rate of structural change.**

   **Theorem 1** *Let $\overline{d\mathcal{A}} = \frac{1}{T}\sum_t d\mathcal{A}_t, \overline{d\mathcal{A}}' = \frac{1}{T}\sum_t d\mathcal{A}'_t$. Then,*

   $$\overline{d\mathcal{A}}' \leq \alpha\overline{d\mathcal{A}} + \beta, \tag{3}$$

   *for some constants $\alpha, \beta \in \mathbb{R}_{\geq 0}$.*

   **Proof.** By the definition of the perturbation matrix $\mathbf{S}_\tau$, $d\mathcal{A}'_t = \|\mathcal{A}'_t - \mathcal{A}'_{t-1}\| = \|(\mathcal{A}_t + \mathbf{S}_t) - (\mathcal{A}_{t-1} + \mathbf{S}_{t-1})\|$. Then, using triangle inequality, we get $d\mathcal{A}'_t \leq \|\mathcal{A}_t + \mathbf{S}_t\| + \|\mathcal{A}_{t-1} + \mathbf{S}_{t-1}\|$. Again using triangle inequality, $d\mathcal{A}'_t \leq \|\mathcal{A}_t\| + \|\mathbf{S}_t\| + \|\mathcal{A}_{t-1}\| + \|\mathbf{S}_{t-1}\|$. Now, since $\|\mathbf{S}_\tau\| \leq \epsilon d\mathcal{A}_t$ and $\|\mathcal{A}_\tau\|$ is a constant, we get $d\mathcal{A}'_t \leq \epsilon d\mathcal{A}_t + \epsilon d\mathcal{A}_{t-1} + C$, for some constant $C$.
   $\overline{d\mathcal{A}}' = \frac{1}{T}\sum_t d\mathcal{A}'_t \leq \frac{1}{T}\sum_t(\epsilon d\mathcal{A}_t + \epsilon d\mathcal{A}_{t-1} + C) \leq 2\epsilon\frac{1}{T}\sum_t d\mathcal{A}_t + C$. Hence, we get that $\overline{d\mathcal{A}}' \leq \alpha\overline{d\mathcal{A}} + \beta$, for some constants $\alpha, \beta \geq 0$. ∎

2. **Perturbations under TDAP constraint preserves the rate of embedding change.**

   **Theorem 2** *Let $d\mathbf{Z}_t = \|\mathbf{Z}_t - \mathbf{Z}_{t-1}\|_1$. Then, for some constants $\gamma, \delta \in \mathbb{R}_{\geq 0}$,*

   $$d\mathbf{Z}' \leq \gamma d\mathbf{Z} + \delta, \tag{4}$$

   **Proof.** Note that $\mathbf{Z}_t = f(\mathcal{A}_t, \mathcal{A}_{t-1}, \cdots, \mathcal{A}_1)$. We consider a stacked vector of flattened matrices $\forall \tau \in [0,t] : \mathbf{q}_{\leq\tau} = (\mathbf{q}_\tau, \mathbf{q}_{\tau-1}, \cdots, \mathbf{q}_1, \mathbf{0}, \mathbf{0}, \cdots, \mathbf{0})$, where $\mathbf{q}_i$ is the flattened vector of $\mathcal{A}_\tau$ and we append $(t - \tau)$ $\mathbf{0}$s to make all vectors $\mathbf{q}_{\leq i}$ of fixed dimension $t$. Then, by Cauchy's Mean Value Theorem in several variables, we have $\mathbf{Z}_t - \mathbf{Z}_{t-1} \leq \nabla f \cdot (\mathbf{q}_{\leq t} - \mathbf{q}_{\leq t-1})$, which gives us $\|\mathbf{Z}_t - \mathbf{Z}_{t-1}\| \leq \|\nabla f\| \|\mathbf{q}_{\leq t} - \mathbf{q}_{\leq t-1}\|$ by Cauchy-Schwarz inequality. We note that $\|\mathbf{q}_{\leq t} - \mathbf{q}_{\leq t-1}\|_1 = \|(\mathbf{q}_t - \mathbf{q}_{t-1}, \cdots, \mathbf{q}_2 - \mathbf{q}_1, \mathbf{q}_1)\|_1 = \sum_t\|\mathbf{q}_t - \mathbf{q}_{t-1}\|_1 = T\overline{d\mathcal{A}}$. Thus, we have $\|\mathbf{Z}_t - \mathbf{Z}_{t-1}\|_1 \leq C\overline{d\mathcal{A}}$ for some constant $C \geq 0$. Using Theorem 1, we get $d\mathbf{Z}' \leq C\overline{d\mathcal{A}}' \leq A\overline{d\mathcal{A}} + B$, for $A = C\alpha, B = C\beta$. By mean-value theorem, we also have $d\mathbf{Z}_t \geq \|\nabla f_{t-1}\| \|\mathbf{q}_{\leq t} - \mathbf{q}_{\leq t-1}\|\cos(\theta) = C_2\overline{d\mathcal{A}}$ ($\theta$ is the angle between $\mathbf{q}_{\leq t}$ and $\mathbf{q}_{\leq t-1}$). Hence, $d\mathbf{Z}' \leq \gamma d\mathbf{Z} + \delta$ for some constants $\gamma, \delta \geq 0$. ∎

We analyze the constants in these bounds in further detail in Appendix C.

## 3.2 ATTACK METHODS UNDER TDAP CONSTRAINT

While the TDAP constraint allows us to limit the effect of the perturbations on the graph's evolution, it is not clear how one can efficiently find perturbations that maximize a loss function under this constraint. To this end, we present two algorithms to solve the optimization problem of Equation 2.

**Greedy.** A greedy strategy can be adopted to find effective perturbations under our TDAP constraint. In this approach, perturbations are selected in a greedy manner based on their gradient values with respect to the downstream loss. However, this does not scale well as one needs to find gradient values corresponding to all the perturbations, which would be $O(T|\mathcal{V}|)$, where $\mathcal{V}$ denotes the set of nodes. Thus, inspired by (Chen et al., 2021b), we find the perturbations in two steps — first, we find the top-gradient perturbation at each time step and then, select the one that reduces the prediction probability the most. In particular, we greedily select the perturbations with the lowest probability such that TDAP($\epsilon$) is not violated for any time-step. We defer the full algorithm to Appendix A.

**Temporal Dynamics-aware Projected Gradient Descent (TD-PGD).** Since the constrained optimization in Equation 2 has a general continuous objective, a greedy approach is only sub-optimal (even for a simpler convex objective) with no theoretical guarantees. A more standard approach to do optimization under a convex constraint is to use projected gradient descent (PGD) (Boyd et al., 2004; Bubeck et al., 2015). Since our problem is in discrete-space, we first relax it into continuous space, find the solution using PGD and then, randomly round it to obtain a valid solution for the discrete problem. In particular, we relax the perturbation matrix $\mathbf{S}_t$ into a continuous vector $\mathbf{s}_t$ and show that a closed-form projection operator exists for the TDAP($\epsilon$) constraint. Algorithm 1 demonstrates the steps involved in this approach (TD-PGD), following the result of Theorem 3.

**Theorem 3** *Suppose $\mathcal{S}$ denotes the feasible perturbation space for the constraints $\|\mathcal{A}'_t - \mathcal{A}_t\|/\|\mathcal{A}_t - \mathcal{A}_{t-1}\| \leq \epsilon$ for all $1 < t < T$ and $\|\mathcal{A}'_1 - \mathcal{A}_1\| \leq \varepsilon_1$. Then, one can project a vector $\mathbf{a}_t$ onto $\mathcal{S}$ using the following projection operator:*

$$\Pi_\mathcal{S}(\mathbf{a}_t) = \begin{cases} P_{[0,1]}(\mathbf{a}_t - \mu_t) & \text{if } \exists\mu_t > 0 : \mathbf{1}^T P_{[0,1]}(\mathbf{a}_t - \mu_t) = \varepsilon_t \\ P_{[0,1]}(\mathbf{a}_t) & \text{if } \mathbf{1}^T P_{[0,1]}(\mathbf{a}_t) \leq \varepsilon_t \end{cases} \tag{5}$$

---

**Algorithm 1** Temporal Dynamics-aware Projected Gradient Descent

---

**Require:** TDAP variables $\varepsilon_t$ (from Thm. 3), Initial vector $\mathbf{s}^{(0)}$, Loss function $\mathcal{L}_{task}$, Actual labels $y_{task}$, Target entities $E_{tg}$, Time steps $T$, Learning rate $\eta_i$, Iterations $N$, Rounding iterations $N_r$

**Ensure:** Perturbation vector $\mathbf{s}^{(i)}$ preserves TDAP($\epsilon$) at every time step $t$

1: **for** $i = 1$ to $N$ **do**
2:     **Gradient descent**: $\mathbf{a}^{(i)} = \mathbf{s}^{(i-1)} + \eta_i \nabla \mathcal{L}_{task}(\{\mathcal{G}_t \oplus \mathbf{s}_t^{(i-1)}; \ \forall t\}, y_{task}, E_{tg})$
3:     **Projection**: For all $t \in [1, T-1]$: $\mathbf{s}_t^{(i)} = \Pi_{\mathcal{S}}(\mathbf{a}_t^{(i)})$ according to Equation 5
4: $\mathbf{S}_t \leftarrow \text{ROUND}(\mathbf{s}_t^{(N)}, N_r, \{\varepsilon_t\})$ from Algorithm 4.

---

*where $\varepsilon_t = \epsilon d\mathcal{A}_t = \epsilon \|\mathcal{A}_t - \mathcal{A}_{t-1}\|$ for $t > 1$, and $P_{[0,1]}(x) = x$ if $x \in [0,1]$, $0$ if $x < 0$, and $1$ if $x > 1$.*

**Proof.** Please see Appendix B for the proof. ∎

Following (Xu et al., 2019b), we use the bisection method (Boyd et al., 2004) to solve the equation $\mathbf{1}^T P_{[0,1]}(\mathbf{a}_t - \mu_t) = \varepsilon_t$ in $\mu_t$ for $\mu_t \in [\min(\mathbf{a}_t - 1), \max(\mathbf{a}_t)]$. This converges in the logarithmic rate, i.e. it takes $O(\log_2[(\max(\mathbf{a}_t) - \min(\mathbf{a}_t - 1))/\xi])$ time for $\xi$-error tolerance.

### 3.3 ONLINE ADVERSARIAL ATTACKS

We also consider the online version of the problem in Equation 2. In this setting, the perturbations are added in real-time, i.e. they are both **immediate** and **irrevocable**. More formally, Equation 2 must now be solved considering online updates of the optimization variables, i.e., (1) $\mathcal{A}'_t$ is updated at time step $t$ without any knowledge of $\mathcal{A}_{t+1:T}$ and (2) $\mathcal{A}'_t$ remains unchanged for future time steps. Note that TDAP constraint must still hold for $\mathcal{A}'_t$ at all time steps $t$.

**Online TD-PGD.** Inspired from its theoretical guarantees in online convex optimization (Zinkevich, 2003), we use Online Projected Gradient Descent for our problem. In this framework, we are given a function $f_t$ for each step $t$ and the goal is to choose $x_t$ in an online manner such that the regret on the offline optimum $x_t^*$, $\mathcal{R}(f, x) := \sum_t (f_t(x_t) - f_t(x_t^*))$ is minimized. In our problem, as defined in Equation 2, we need to minimize a loss $h(\mathcal{A}_{1:T-1})$ at the final time step $T$. To use online gradient descent, we thus need to write $h$ as $\sum_t f_t(\mathcal{A}_t)$ for some $f_t$. Let us assume that $h$ is a cross-entropy loss and that the embeddings at each time $t$ are encoded in a sequential manner. Then,

$$h(\{\mathcal{A}_t\}_{t=1}^{T-1}) = -\sum_{d \in \mathcal{D}} y(d) \log p(d, \{\mathcal{A}_t\}) = \sum_{d \in \mathcal{D}} y(d) \sum_t -\log p(d, \mathcal{A}_t | \mathcal{A}_1, \cdots, \mathcal{A}_{t-1})$$
$$= \sum_t -\sum_{d \in \mathcal{D}} y(d) \log p(d, \mathcal{A}_t | \mathcal{A}_1, \cdots, \mathcal{A}_{t-1}).$$

Thus, we can define $f_t(\mathcal{A}_t) = -\sum_{d \in \mathcal{D}} y(d) \log p(d, \mathcal{A}_t | \mathcal{A}_1, \cdots, \mathcal{A}_{t-1})$, which is the prediction loss for the data points $\mathcal{D}$ at time step $t$. Algorithm 1 can then be updated to find attacks in real time, following Online Gradient Descent. In particular, for time $t$, we find perturbations $\mathbf{s}_t^{(i)}$ by replacing the loss in line 2 with $\mathcal{L}(\{\mathcal{G}_\tau \oplus \mathbf{s}_\tau^{(i-1)}\}_{\tau=1}^t, y(t), \cdot)$. Since the projection operator for TDAP (Equation 5) depends only on the current time step $t$, we can independently project for the current time, i.e. line 3 remains $\mathbf{s}_t^{(i)} = \Pi_{\mathcal{S}}(\mathbf{a}_t^{(i)})$. For more details about the algorithm, refer Appendix D.

## 4 EXPERIMENTAL SETUP

**Datasets.** We use 3 datasets for dynamic link prediction — Radoslaw[1], UCI [1], and Reddit[2]. Radoslaw and UCI are email communication networks, where two nodes (users) are connected if they have an email communication at time $t$. Reddit is a hyperlink network representing directed connections between subreddits if there is a hyperlink from one to the other at a given timestamp (Kumar et al., 2018). For node classification task, we use one publicly-available dataset, DBLP-5

---

[1]http://konect.cc/networks/
[2]https://snap.stanford.edu/data/soc-RedditHyperlinks.html

(Xu et al., 2019a). This is a co-author network with node attributes as word2vec representations of the author's papers. There are 5 node labels representing the different fields that the authors belong to. Please refer to Appendix E for more details regarding the datasets.

**Attack Methods.** We consider 4 different attack methods to find perturbations under TDAP constraint. **(1) TD-PGD** is a projected gradient descent with a valid projection operator for the TDAP constraint, as specified in Algorithm 1. **(2) TGA**($\epsilon$) greedily selects the perturbation with the highest gradient value of the loss (we adapt TGA (Chen et al., 2021b) to our setting, as specified in Section 3.2). **(3) DEGREE** flips the edges (adds or deletes if already there) attached with the highest degree nodes in the graph at each time step, while making at most $\epsilon d \mathcal{A}_t$ perturbations. **(4) RANDOM** randomly flips (add or delete) at most $\epsilon d \mathcal{A}_t$ edges at each time step $t$.

**Victim Models.** We test the performance of the above attack methods on 3 different discrete-time dynamic graph models. **(1) GC-LSTM** (Chen et al., 2018) embeds GCN into an LSTM to encode the sequence of graphs. **(2) EVOLVEGCN** (Pareja et al., 2020) uses a recurrent model (RNN-LSTM) to evolve the weights of a GCN. We use the EVOLVEGCN-O version for our experiments. **(3) DYSAT** (Sankar et al., 2020) utilizes joint structural and temporal self-attention to embed.

**Metrics.** We use the relative drop, as defined below, to evaluate the efficacy of the attack methods.

$$\text{Rel. Drop (\%)} = \frac{\text{Perturbed performance} - \text{Original performance}}{\text{Original performance}} \times 100, \qquad (6)$$

where performance is evaluated using ROC-AUC for dynamic link prediction and using Accuracy for node classification.

In order to evaluate the detectability of the attack methods, we propose a novel metric **Embedding Variability (EV)** to compare the consecutive embedding difference for the perturbed graph and that for the original graph. Consecutive embedding difference has been used to identify anomalies in the data (Goyal et al., 2018). Here, we measure how the range of this difference changes due to the perturbation. In particular, we consider

$$EV(\mathbf{Z}, \mathbf{Z}') := \left| 1 - \frac{\max_\tau d\mathbf{Z}'_\tau - \min_\tau d\mathbf{Z}'_\tau}{\max_\tau d\mathbf{Z}_\tau - \min_\tau d\mathbf{Z}_\tau} \right| \qquad (7)$$

This measures the relative variability of the consecutive change in the embedding space. For the attacks to be less detectable, this metric should be close to 0. Note that we do not directly optimize for this metric in our constraint.

## 5 RESULTS

We compare the performance of different attack methods for the 3 victim models on link prediction and node classification tasks. We also test the attack performance on dynamic link prediction in the novel online setting. For all the experiments, we vary $\epsilon$ from 0 to 1 and fix $\varepsilon_1 = \min_{t>1} \varepsilon_t := \epsilon d \mathcal{A}_t$.

### 5.1 DYNAMIC LINK PREDICTION

In this section, we show the attack performance on the task of dynamic link prediction (Sankar et al., 2020; Pareja et al., 2020). The task here is to predict whether a link $(u, v)$ will appear or not at the future timestep. The objective of the attacker, thus, is to introduce perturbations in the past time steps to make the model mispredict link's existence in future. We test the victim models on the final snapshot for a set of target links. We consider 3 different sets of 100 positive and 100 negative random targets and show the mean relative ROC-AUC drop with error bars.

Figure 1 shows the performance of different attack methods on this task across different datasets and models. TD-PGD outperforms the other baselines in all cases, except in GC-LSTM model trained on `Reddit`. Moreover, TD-PGD is able to drop the AUC by up to 4 times the baselines and lead to $\sim 100\%$ drop in the AUC, completely flipping the prediction. We also note that TD-PGD often has a continuously decreasing slope and its performance saturates much later than the other baselines. The second best baseline is often TGA($\epsilon$) but in many cases, it is only as good as random. One can also note that EVOLVEGCN shows a larger drop than the other 2 models across all datasets. This may pertain to the lower model complexity of EVOLVEGCN compared to others. We discuss these results in further detail in Appendix H.

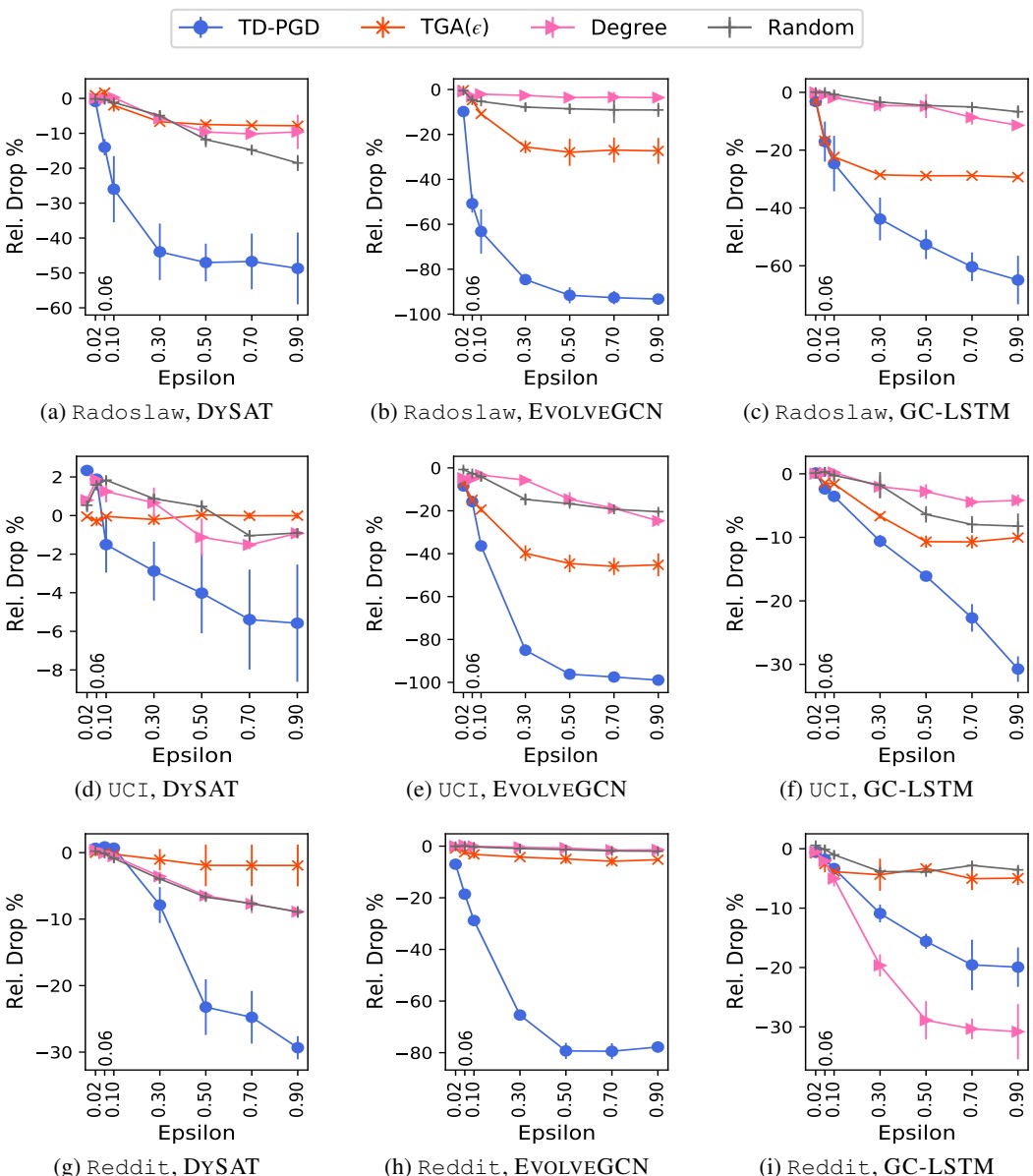

Figure 1: Attack performance on dynamic link prediction task across datasets and models.

**Detectability of the attack methods:** Here, we use the EV metric, as defined in Section 4, to assess how detectable the attack methods are under our constraint. Table 2 compares the attack performance of the best method TD-PGD at $\epsilon = 0.5$ and the corresponding variability in the embeddings, as given by Equation 7. One can note that $E$ is smaller than 0.5 in all but one case and that TD-PGD is able to cause up to 90% drop in performance while changing the evolution by only a factor of 0.22, on average. This shows that TDAP allows for undetectable yet effective attacks. Please refer to Appendix H for more analysis.

## 5.2 NODE CLASSIFICATION

In this section, we compare the attack performance on the task of semi-supervised node classification (Pareja et al., 2020). In this task, the objective is to predict node labels of a set of nodes while knowing the labels of the other nodes at that time step.

| Dataset | Model | Rel. Drop % | EV |
|---|---|---|---|
| Radoslaw | DYSAT | 47.03 (5.42) | 0.04 (0.18) |
| | EVOLVEGCN | 91.61 (3.58) | 0.45 (1.29) |
| | GC-LSTM | 52.63 (5.09) | 0.18 (1.16) |
| UCI | DYSAT | 4.02 (2.08) | 0.14 (0.25) |
| | EVOLVEGCN | 96.21 (0.17) | 0.22 (0.35) |
| | GC-LSTM | 16.12 (0.75) | 0.35 (0.47) |
| Reddit | DYSAT | 23.24 (4.18) | 0.14 (0.45) |
| | EVOLVEGCN | 79.31 (3.13) | 0.81 (6.37) |
| | GC-LSTM | 15.59 (1.28) | 0.36 (0.59) |

Table 2: Comparison of attack performance and detectability (refer Section 4) for TD-PGD at $\epsilon = 0.5$. Mean values are noted with standard deviations in the parentheses.

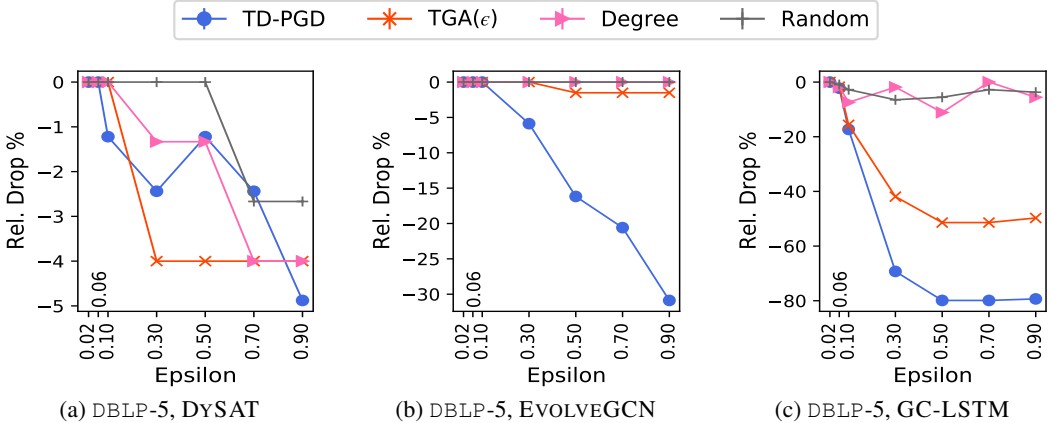

(a) DBLP-5, DYSAT    (b) DBLP-5, EVOLVEGCN    (c) DBLP-5, GC-LSTM

Figure 2: Attack performance on node classification task for top degree nodes across models.

Figure 2 shows the effect of structural perturbations on node classification task by different attack strategies for the 3 models. Misclassifying the labels for influential top-degree targets can significantly impact a model's usability in practice. Therefore, we consider the performance on 50 top-degree nodes for each class. Results show that TD-PGD outperforms the baselines in all models except DYSAT, in which all attacks perform almost equally. In particular, TD-PGD is able to cause a 30% drop in EVOLVEGCN while the baselines only lead to a drop of 5%. We show the performance on random targets in Appendix H and note that feature perturbations are more effective in these cases when the structural information is sparse.

## 5.3 ONLINE ADVERSARIAL ATTACKS

In this section, we consider the online setting as described in Section 3.3 and compare the online version of TD-PGD with the RANDOM and DEGREE baselines on the dynamic link prediction task. Since the loss at the final step is not available at time step $t$, one cannot select the perturbations in a greedy manner of the gradients. Therefore, we do not have a TGA($\epsilon$) baseline for this setting.

Figure 3 shows the average performance of the three methods for dynamic link prediction task on 3 datasets over 3 random seeds. Please refer Appendix H for results on node classification. TD-PGD outperforms the other online baselines in most cases and is able to achieve competent performance to the offline version. In particular, it shows up to 5 times improvement over the existing baselines (for EVOLVEGCN on UCI), which is close to the offline TD-PGD as shown in Figure 1. However, TD-PGD does not perform well in Figures 3a and 3i. Online TD-PGD perturbs the graph at time $t$ according to the loss at that time step rather than the final step. While it is guaranteed to give strong bounds for a convex objective, some models may learn a complex non-convex function in its input. We conjecture that the degradation may be due to such functions being learned in these cases.

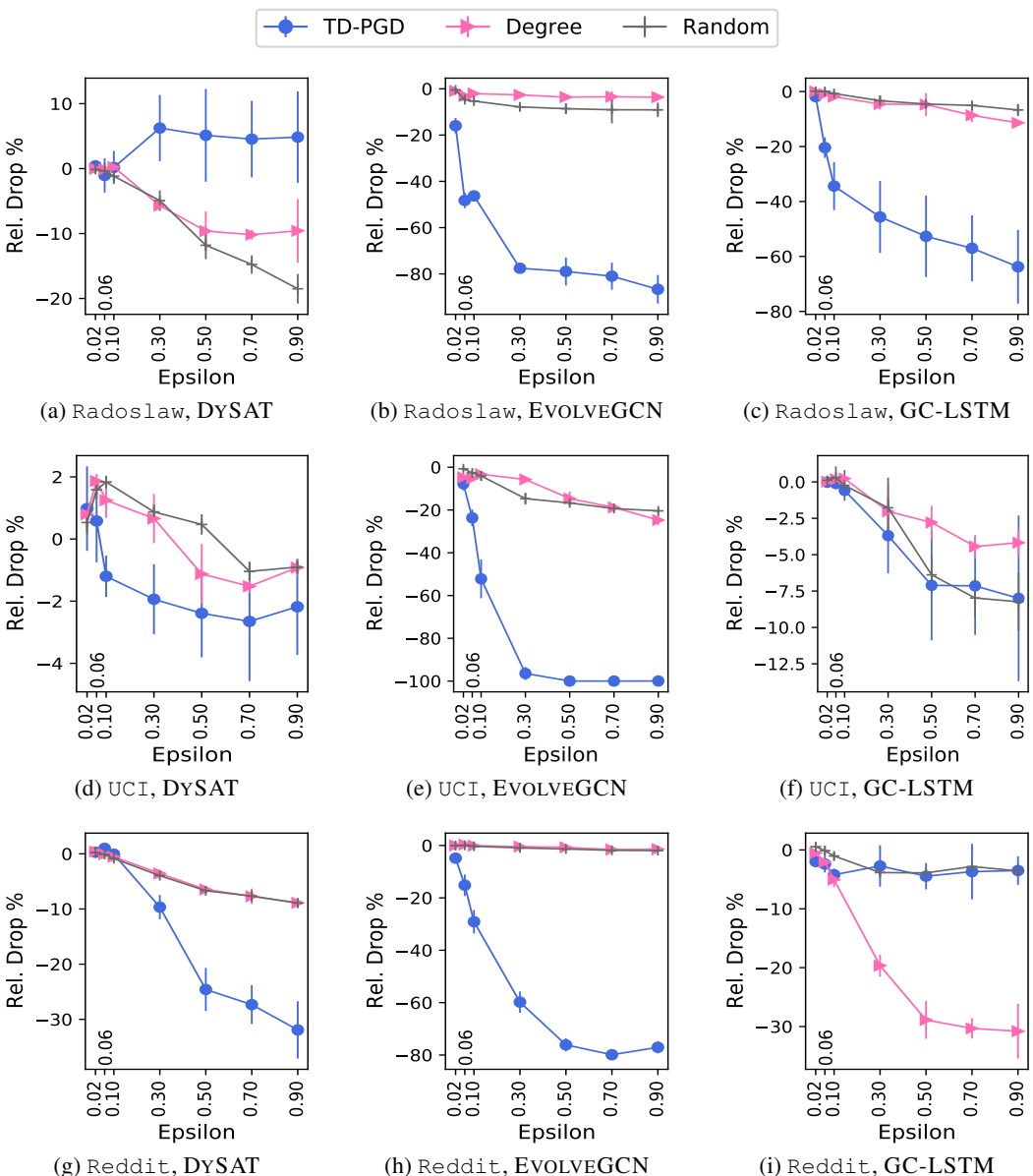

Figure 3: Online Attack performance on dynamic link prediction task across datasets and models.

## 6 CONCLUSION

Our work has shown that state-of-the-art dynamic graph models can be effectively attacked while preserving the temporal dynamics. We introduce a novel Temporal Dynamics-Aware Perturbation (TDAP) constraint to devise perturbations in discrete-time dynamic graphs that preserves the graph evolution in both the structural and embedding spaces. Next, we present an effective PGD-based approach to find perturbations under this constraint and show improved attack performance than baselines in both offline and online settings. We hope that our work serves as a first step towards opening exciting research avenues on studying attacks and defense mechanisms for both discrete and continuous-time dynamic graphs. Some limitations of our current exposition can be noted. First, the proposed method TD-PGD is not memory-efficient and may not scale to larger graphs (more details in Appendix G). Second, randomly rounding the solution to discrete space may lead to suboptimal perturbations. Future work can study more effective and efficient methods to attack dynamic graphs under TDAP constraint in the more restrictive black-box setting.

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

APPENDIX

## A  GREEDY APPROACH

---

**Algorithm 2** Greedy Algorithm (TGA($\epsilon$))

---

**Require:** TDAP variables $\varepsilon_t$ (from Equation 5), Initial perturbation vector $\mathbf{s}^{(0)}$, Loss function $\mathcal{L}_{task}$,
  Probability function $p_{\mathcal{M}}$ predicting link existence probability, actual labels $y_{task}$, Target entities
  $E_{tg}$, Time steps $T$.
1: For all $t$: $\mathcal{G}'_t \leftarrow \mathcal{G}_t$. Attack history $\mathbf{H} \leftarrow \phi$.
2: **while** True **do**
3:   **for** t=1 to T **do**
4:     $\mathbf{s}_t \leftarrow \phi$
5:     **for** $n_{tg}$ in $E_{tg}$ **do**
6:       **grads**$[v] \leftarrow \partial\mathcal{L}_{task}(\mathcal{G}'_{1:T-1})/\partial(n_{tg}, v)$ for $v$ in $\mathcal{V}$.
7:       Pick the first $v$ in the descending order of **grads** such that $(n_{tg}, v) \notin \mathbf{S}$.
8:       $\mathbf{s}_t$.append($(n_{tg}, v)$).
9:       **probs**.append($p_{\mathcal{M}}(\mathcal{G}'_t \oplus \mathbf{s}_t)$)
10:     Pick $\mathbf{s}_\tau$ in the descending order of **probs** such that $\|\mathbf{H}_t \oplus \mathbf{s}_\tau\| \leq \varepsilon_t$.
11:     **if** no $\tau$ found **then**
12:       break
13:     $\mathcal{G}'_t \leftarrow \mathcal{G}_t \oplus \mathbf{s}_t$.
14:     $\mathbf{H}_t \leftarrow \mathbf{H}_t \oplus \mathbf{s}_t$.

---

## B  PROOF OF THEOREM 3

Suppose $\mathcal{S}$ denotes the feasible perturbation space for the constraints $\|\mathcal{A}'_t - \mathcal{A}_t\|/\|\mathcal{A}_t - \mathcal{A}_{t-1}\| \leq \epsilon$
for all $1 < t < T$ and $\|\mathcal{A}'_1 - \mathcal{A}_1\| \leq \varepsilon_1$. Then, one can project a vector $\mathbf{a}_t$ onto $\mathcal{S}$ using the following
projection operator:

$$\Pi_{\mathcal{S}}(\mathbf{a}_t) = \begin{cases} P_{[0,1]}(\mathbf{a}_t - \mu_t) & \text{if} \exists \mu_t > 0 : \mathbf{1}^T P_{[0,1]}(\mathbf{a}_t - \mu_t) = \varepsilon_t \\ P_{[0,1]}(\mathbf{a}_t) & \text{if } \mathbf{1}^T P_{[0,1]}(\mathbf{a}_t) \leq \varepsilon_t \end{cases} \tag{8}$$

where $\varepsilon_t = \epsilon d\mathcal{A}_t = \epsilon\|\mathcal{A}_t - \mathcal{A}_{t-1}\|$ for $t > 1$, and $P_{[0,1]}(x) = x$ if $x \in [0,1]$, 0 if $x < 0$, and 1 if $x > 1$.

**Proof.** Let the perturbation vector be $\mathbf{s} = [\mathbf{s}_1, \mathbf{s}_2, \cdots, \mathbf{s}_{T-1}]^T$. Then, for all $t \in [2, T-1]$, since
$\mathbf{s}_{t_t} = \mathcal{A}_t - \mathcal{A}_{t-1}$, our constraint becomes $\|\mathbf{s}_t\| \leq \epsilon d\mathcal{A}_t \leq \epsilon\|\mathcal{A}_t - \mathcal{A}_{t-1}\| =: \varepsilon_t$, which reduces to
$\mathbf{1}^T \mathbf{s}_t \leq \varepsilon_t$. For $t = 1$, we have $\|\mathcal{A}'_1(\mathcal{V}_1) - \mathcal{A}_1(\mathcal{V}_1)\| \leq \varepsilon_1$, which also becomes $\mathbf{1}^T \mathbf{s}_1 \leq \varepsilon_1$. Hence, we
have the constraint $\mathbf{1}^T \mathbf{s}_t \leq \varepsilon_t$ for all $t$.

By definition, then, projection operator must be $\Pi_{\mathcal{S}}(\mathbf{a}) = \arg\min_{\mathbf{s}\in\mathcal{S}} \frac{1}{2}\|\mathbf{s} - \mathbf{a}\|^2$, where $\mathcal{S} = \{\mathbf{s} \in [0,1]^n \mid \mathbf{1}^T \mathbf{s}_t \leq \varepsilon_t \ \forall t\}$. This reduces to the following optimization problem:

$$\Pi_{\mathcal{S}}(\mathbf{a}) = \arg\min_{\mathbf{s}\in\mathcal{S}} \frac{1}{2} \sum_{t\in[1,T]} \|\mathbf{s}_t - \mathbf{a}_t\|^2 + \mathcal{I}_{[0,1]}(\mathbf{s}_t), \tag{9}$$

$$\text{such that} \quad \forall t \in [1, T] : \mathbf{1}^T \mathbf{s}_t \leq \varepsilon_t$$

where $\mathcal{I}_{[0,1]}(x) = 0$ if $x \in [0,1]^n$ and $\infty$ otherwise.

This can be solved by the Lagrangian method. We note that the Lagrangian function of the above
optimization problem is $\mathcal{L}(\mathbf{s}, \mathbf{a}, \mu) = \sum_{t\in[1,T]} \left(\frac{1}{2}\|\mathbf{s}_t - \mathbf{a}_t\|^2 + \mathcal{I}_{[0,1]}(\mathbf{s}_t) + \mu_t(\mathbf{1}^T \mathbf{s}_t - \varepsilon_t)\right)$.

$\partial\mathcal{L}/\partial\mathbf{s}_t = 0 \implies \mathbf{s}_t = \mathbf{a}_t - \mu_t$. However, if $s_{t,i} < 0$ or $s_{t,i} > 1$ for any $i$, then $\mathcal{L} = \infty$. Thus, the
minimizer to the above function is $\mathbf{s}^*_t = P_{[0,1]}(\mathbf{a}_t - \mu_t)$, where $P_{[0,1]}(x) = x$ if $x \in [0,1]$, 0 if $x < 0$ and
1 if $x > 1$. In addition, the solution must satisfy the following KKT conditions $\forall t$:

(1) $\mu_t(\mathbf{1}^T \mathbf{s}_t^* - \varepsilon_t) = 0$,   (2) $\mu_t \geq 0$,   (3) $\mathbf{1}^T \mathbf{s}_t^* \leq \varepsilon_t$.

If $\mu_t > 0$, then we must have $\mathbf{1}^T P_{[0,1]}(\mathbf{a}_t - \mu_t) = \varepsilon_t$. Otherwise if $\mu_t = 0$, then $\mathbf{1}^T P_{[0,1]}(\mathbf{a}_t) \leq \varepsilon_t$.

■

## C   Interpreting The Theoretical Bounds On Evolution

In this section, we analyze the constants in the theoretical bound on the average structural change after perturbation under TDAP constraint, as proved in Theorem 1. In particular, we show how Theorems 1 and 2 shows that a local TDAP constraint preserves the average trend of structural and embedding change as a linear combination of the original trends. Thus, if the linear factors are small enough, then, the average trend of evolution remains well-preserved. This ensures imperceptibility as perturbations that lead to a change in the trend of structural/embedding evolution can be easily detected by various anomaly detection algorithms (Cai et al., 2021; Goyal et al., 2018). In particular, while the TDAP constraint allows us to find a bound on the perturbed trend in terms of the original trend, no existing constraints give us such a bound. This is because we constrain the number of perturbations introduced at each time step to be within a factor of the changes introduced in the original graph at that time step while existing constraints only impose a global constraint on the perturbations over all time steps.

One can note that $\alpha = 2\epsilon$ and $\beta = \frac{1}{T}\sum_t \|\mathcal{A}_t\| + \|\mathcal{A}_{t-1}\| = \frac{2}{T}\sum_t \|\mathcal{A}_t\| - \|\mathcal{A}_T\| \leq \frac{2}{T}\sum_t \|\mathcal{A}_t\|$. In other words, the average rate of structural change after perturbations within a TDAP constraint remains within a factor of 2 times the permitted fraction for the TDAP constraint of the original average rate, plus 2 times the average no. of edges in the adjacency matrix. Since we consider a targeted case, the adjacency matrix includes only a single column and the additive factor corresponds to 2 times the average degree of the target vertex over all time steps. Since $\epsilon$ is a parameter that is chosen by the attacker, a small enough $\epsilon$ allows the attacker to preserve the evolution within a small enough multiplicative factor. The additive shift in the trend is equal to twice the average degree of the target vertex, which can inform the choice of the target for the attacker according to his permitted limit.

In comparison, a simple budget constraint gives no bound on the average structural change in terms of the original change. This is because, $\sum_t \|S_t\| \leq \mathcal{B}$, where $\mathcal{B}$ is the budget. Thus, using similar calculations as Theorem 1, we get $d\mathcal{A}_t' \leq \frac{2}{T}\mathcal{B} + \frac{2}{T}\sum_t \|\mathcal{A}_t\|$. Therefore, perturbations introduced using a budget constraint can exceed the average trend of structural change when $\mathcal{B}/T > \epsilon d\mathcal{A}_t$.

## D   Online Gradient Descent

**Problem 1** *Given a convex set $\mathcal{K}$ and at every step $i = 1, 2, \cdots, T$, we are presented with $f_i$ such that we have to choose a solution $x^{(i)} \in \mathcal{K}$ at every step in a way that minimizes the regret $\mathcal{R} = \sum_i (f_i(x^{(i)}) - f_i(x^*))$ without knowing the future functions $f_{i+1:T}$.*

**Lemma 1** *(Zinkevich, 2003) Let G denote an upperbound on $\|\nabla f_i(x)\|_2$ for any $x \in \mathcal{K}$ and any $i$, and let $D = \max_{x,y \in \mathcal{K}} \|x - y\|_2$ be the diameter of $\mathcal{K}$. The online gradient descent algorithm with $\eta = \frac{D}{G\sqrt{T}}$ gives a regret per step of at most $\frac{2DG}{\sqrt{T}}$ after $T$ steps.*

Online Gradient Descent requires replacing $\nabla f(x^{(i)})$ with $\nabla f_i(x^{(i)})$. Algorithm 3 presents the corresponding online version for TD-PGD.

## E   Data statistics and pre-processing

Table 3 shows the statistics of the datasets used in this paper. DBLP-5 is the node classification dataset with 5 labels while the others are dynamic link prediction datasets with varying sizes. The datasets span over differing periods of time. We split each of them into finite number of timesteps to keep a large enough number of time steps while maintaining a realistic period of splitting. Radoslaw is split using a 3-week period in 13 snapshots while the 13 snapshots in UCI denote a

---

**Algorithm 3** Online Projected Gradient Descent for dynamic graphs

---

**Require:** Current time $t$, TDAP variable $\varepsilon_t$, Initial perturbation vector $\mathbf{s}^{(0)}$, Loss function $\mathcal{L}_{task}$,
   Actual labels $y_{task}$, Target entities $E_{tg}$, Learning rate $\eta_i$, Iterations $N$, Rounding iterations $N_r$.

**Ensure:** Perturbation vector $\mathbf{s}_t^{(i)}$ preserves TDAP($\epsilon$)
 1: **for** $i = 1$ to $N$ **do**
 2:     **Gradient descent**: $\mathbf{a}_t^{(i)} = \mathbf{s}_t^{(i-1)} + \eta_i \nabla \mathcal{L}_{task}(\{\mathcal{G}_\tau \oplus \mathbf{s}_\tau^{(i-1)}; \ \forall \tau \in [1,t]\}, \ y_{task}(t), \ E_{tg})$
 3:     **Projection**: $\mathbf{s}_t^{(i)} = \Pi_{\mathcal{S}}(\mathbf{a}_t^{(i)})$
 4: $\mathbf{S}_t \leftarrow \texttt{ROUND}\,(\mathbf{s}_t^{(N)}, N_r, \varepsilon_t)$ from Algorithm 4.

---

2-week period. The `Reddit` dataset is spanned over 3 years, thus we use a 2-month split to obtain the 20 snapshots. We use the publicly available pre-processed data for `DBLP`-5 (Xu et al., 2019a).

For datasets with no node features, i.e., `Radoslaw`, `UCI`, and `Reddit`, we use uniformly random features with dimension 10. The pre-processed `DBLP`-5 has 100 node features for each node.

|  | # Nodes | # Edges | # Time-steps | # Labels |
|---|---|---|---|---|
| Radoslaw | 167 | 22K | 13 | - |
| UCI | 1.9K | 24K | 13 | - |
| Reddit | 35K | 715K | 20 | - |
| DBLP-5 | 6.6K | 43K | 10 | 5 |

Table 3: Description of the datasets.

# F ADDITIONAL DETAILS ON EXPERIMENTAL SETUP

## F.1 SETUP

We consider a targeted setting with single targets, that are selected using either a random sampling or a degree-biased sampling. Each target is attacked one-by-one and the total performance of an attacker is measured using either an ROC-AUC or an Accuracy over the set of sampled targets.

## F.2 HYPERPARAMETERS

For TD-PGD optimization, we used ADAM optimizer (Kingma & Ba, 2014) with the initial learning rate of 10. The initial perturbation vector $\mathbf{s}^{(0)}$ was initialized with all ones, thus, giving each perturbation an equal chance at the start. The algorithm was run for 50 iterations. We use Binary Cross Entropy loss for training dynamic link prediction and Weighted Cross Entropy for node classification.

As we show in the Appendix H.6, TGA($\epsilon$) algorithm does not scale well to higher $\epsilon$ and larger perturbation space. For this reason, we stop the greedy search if the time taken exceeds 300 s, which is at least 3 times that of TD-PGD.

## F.3 VICTIM MODEL TRAINING

Table 4 shows the performance on the test set of the victim models on different datasets. The performance is evaluated using ROC-AUC for the dynamic link prediction and using Accuracy for the node classification tasks. The test set for dynamic link prediction task denotes the edges and non-edges in the final snapshot, while for node classification, we use a 20% held-out set of nodes as the test set for noting the performance and attacking.

| Dataset | Model | Perf. |
|---------|-------|-------|
| Radoslaw | DYSAT | 0.743 |
| | EVOLVEGCN | 0.742 |
| | GC-LSTM | 0.813 |
| UCI | DYSAT | 0.952 |
| | EVOLVEGCN | 0.873 |
| | GC-LSTM | 0.968 |
| Reddit | DYSAT | 0.947 |
| | EVOLVEGCN | 0.939 |
| | GC-LSTM | 0.941 |
| DBLP-5 | DYSAT | 0.699 |
| | EVOLVEGCN | 0.687 |
| | GC-LSTM | 0.695 |

Table 4: Performance (Perf.) on test set for different datasets and models. For Radoslaw, UCI, Reddit, we use ROC-AUC as the performance metric and for DBLP-5, Accuracy is used.

### F.4 IMPLEMENTATION

We use the TorchGeometric-Temporal [3] implementation of EVOLVEGCN and GC-LSTM to train these models. For DYSAT, we use the pytorch implementation [4]. We adapt the official code of TGA[5] to implement the greedy approach. For TD-PGD implementation, we adapt the DeepRobust [6] implementation for dynamic graphs under TDAP constraint. Full code is provided in the attached supplementary for reference.

## G COMPLEXITY ANALYSIS

**TGA($\epsilon$):** It makes $O(\epsilon \sum_t d\mathcal{A}_t)$ backward calls to the victim model for gradient calculation. Let the gradient calculation takes $T_{bw}$ for a model $\mathcal{M}$. Then, the total time for the greedy is given by $O(\epsilon \sum_t d\mathcal{A}_t T_{bw})$.

**TD-PGD:** It makes $O(N)$ backward calls to the victim model for gradient calculation. In addition, the projection step takes $O(\sum_t \log_2[(\max(\mathbf{a}_t) - \min(\mathbf{a}_t - 1))/\xi])$ time per iteration and the random rounding takes $O(\sum_t |\mathbf{s}_t|) = O(T|\mathcal{V}|)$ time. Therefore, the total time taken by TD-PGD is given by $O(NT_{bw} + N\sum_t \log_2[(\max(\mathbf{a}_t) - \min(\mathbf{a}_t - 1))/\xi] + T|\mathcal{V}|)$.

Note that the bottleneck, here, is the term $T_{bw}$ and TD-PGD replaces the dependence on the no. of perturbations to a constant (which is fixed to 50 in the experiments for all the datasets). Therefore, while TGA($\epsilon$) finds it hard to scale to larger datasets, TD-PGD can scale as long as there is enough memory. It is worthwhile to note here that TD-PGD would clearly need more memory storage as it needs to store the complete perturbation vector $\mathbf{s}$ in the memory during optimization. A recent paper proposes an alternative method, called PR-BCD, which stores only optimizes for a fixed set of random perturbations (Geisler et al., 2021). Such an approach can be employed with our projection operator to scale to larger datasets at fixed memory usage. We leave testing the effectiveness of this approach on large dynamic graphs for the future.

---

[3] https://pytorch-geometric.readthedocs.io/en/latest/index.html#

[4] https://github.com/FeiGSSS/DySAT_pytorch

[5] https://github.com/jianz94/tga

[6] https://github.com/wenqifan03/RobustTorch

## H    ADDITIONAL EXPERIMENTS

### H.1    PGD PERFORMANCE ON DIFFERENT MODELS

Figure 4 compares the TD-PGD attack performance over random targets of different victim models for dynamic link prediction. One can note that EVOLVEGCN is the least robust among these models as TD-PGD causes the most drop in this model across datasets. On the other hand, GC-LSTM and DYSAT show similar drop with DYSAT being slightly more robust among them. One can explain this result using the architectural differences between these models as DYSAT, being an attention-based architecture, makes use of more parameters than other models. GC-LSTM embeds GCN into an LSTM and thus, uses the number of parameters than a combination of GCN and LSTM. EVOLVEGCN uses a LSTM architecture to evolve the weights of a GCN, which is more efficient.

### H.2    DYNAMIC LINK PREDICTION ON TOP-DEGREE TARGETS

Figure 5 shows the attack performance on dynamic link prediction with targets being 100 top-degree edges and non-edges each. These targets are picked in the decreasing order of the sum of the degrees of the end-nodes over the time-steps. One can note that TD-PGD outperforms the existing baselines in most cases for these targets as well.

### H.3    EMBEDDING VARIABILITY

In this section, we plot the raw values of embedding variability (EV) of different models with increasing $\epsilon$. Figure 6 shows the mean EV values with their standard deviations at different $\epsilon$, while Figure 7 shows the median values with the interpercentile range (10% and 90%). We note that mean and median values are close to zero for all the methods (only DEGREE reaches 1). DEGREE is usually the most detectable of the methods based on the variability caused in the embeddings. Huge variance in Figure 6 can be attributed to the existence of outliers in the data as the median and 90% quantile value is usually low for all methods, except for DEGREE (as shown in Figure 7).

### H.4    NODE CLASSIFICATION OVER RANDOM TARGETS

In this section, we compare the attack performance on the node classification task over randomly-selected target nodes from the test set. In addition to structural perturbations, we also do feature perturbations while following the TDAP constraint.

**Structural perturbations**    Figure 8 compares the drop caused by different attack methods on the node classification task for different models over random targets. One can note that all attack models perform as good as the other on EVOLVEGCN and DYSAT, while TD-PGD outperforms others on

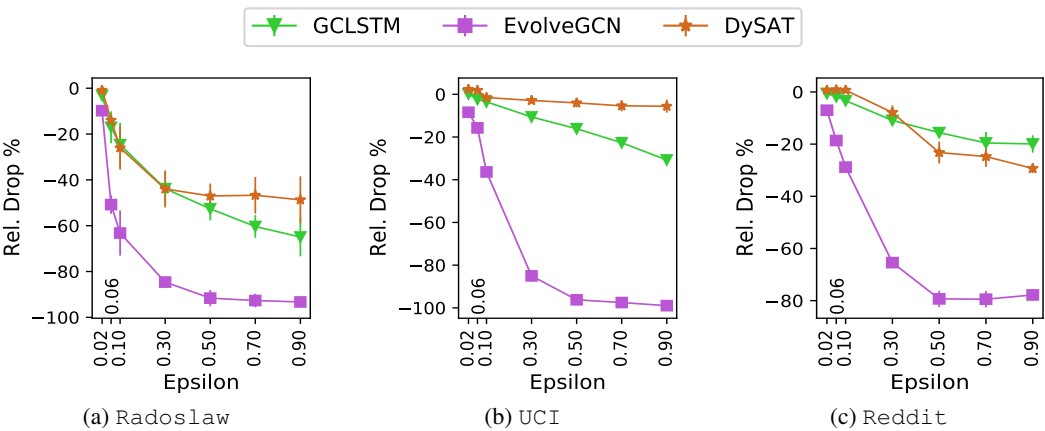

Figure 4: PGD performance on dynamic link prediction task across datasets and models.

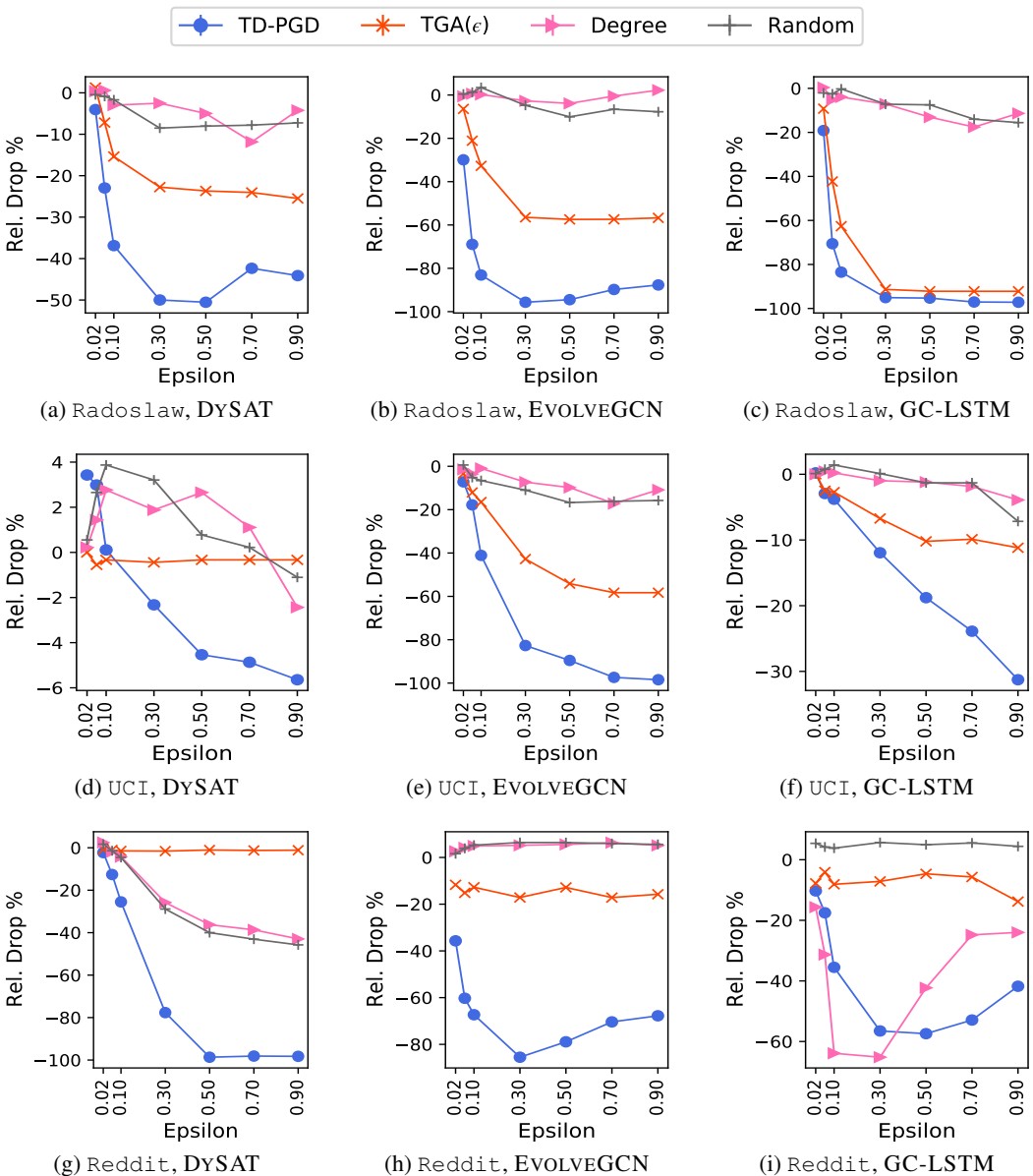

Figure 5: Attack performance on dynamic link prediction task across datasets and models for top-degree targets.

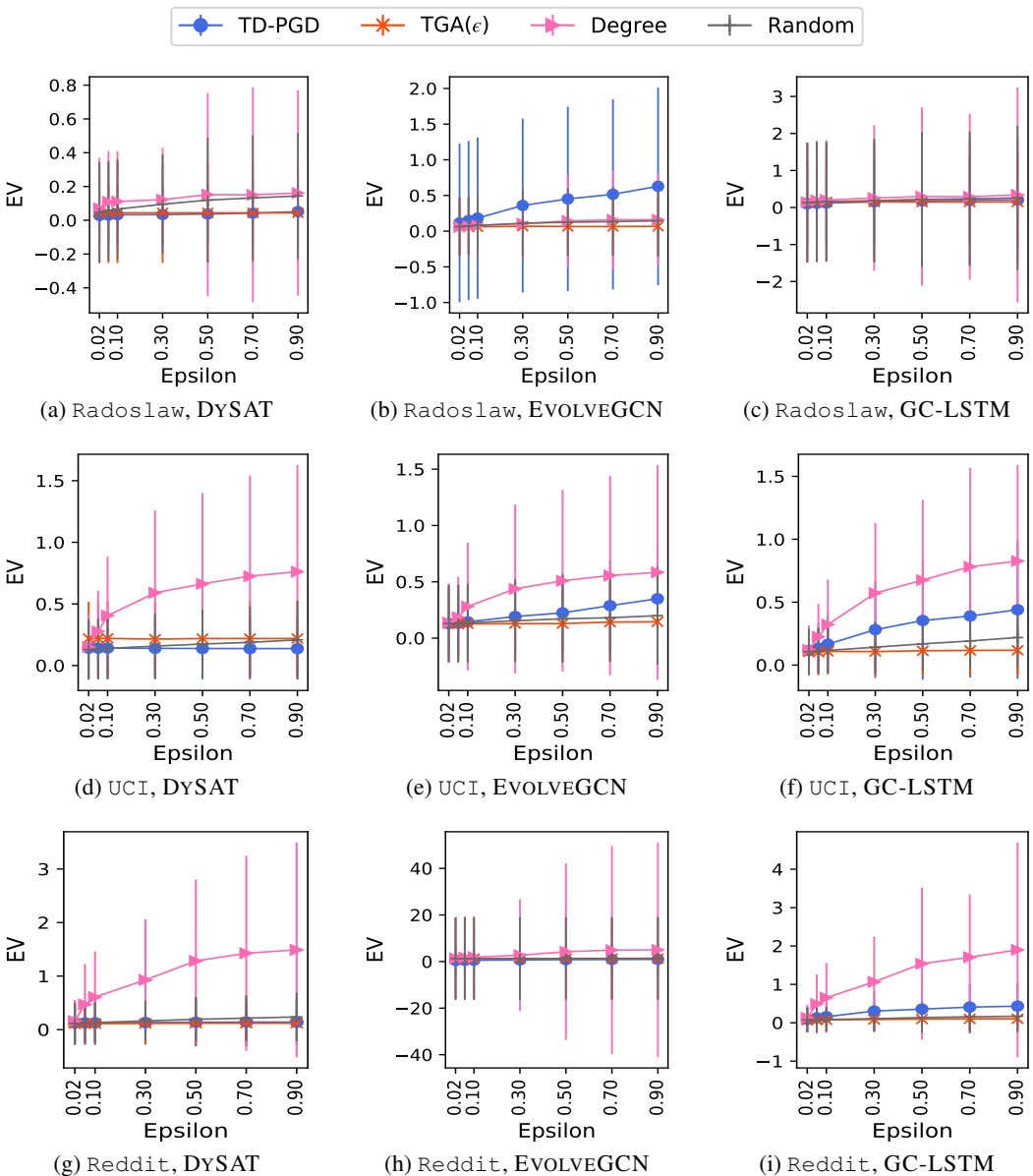

Figure 6: Comparison of embedding variability (EV) on link prediction task across datasets and models. We show mean values for each method with standard deviation as error bars.

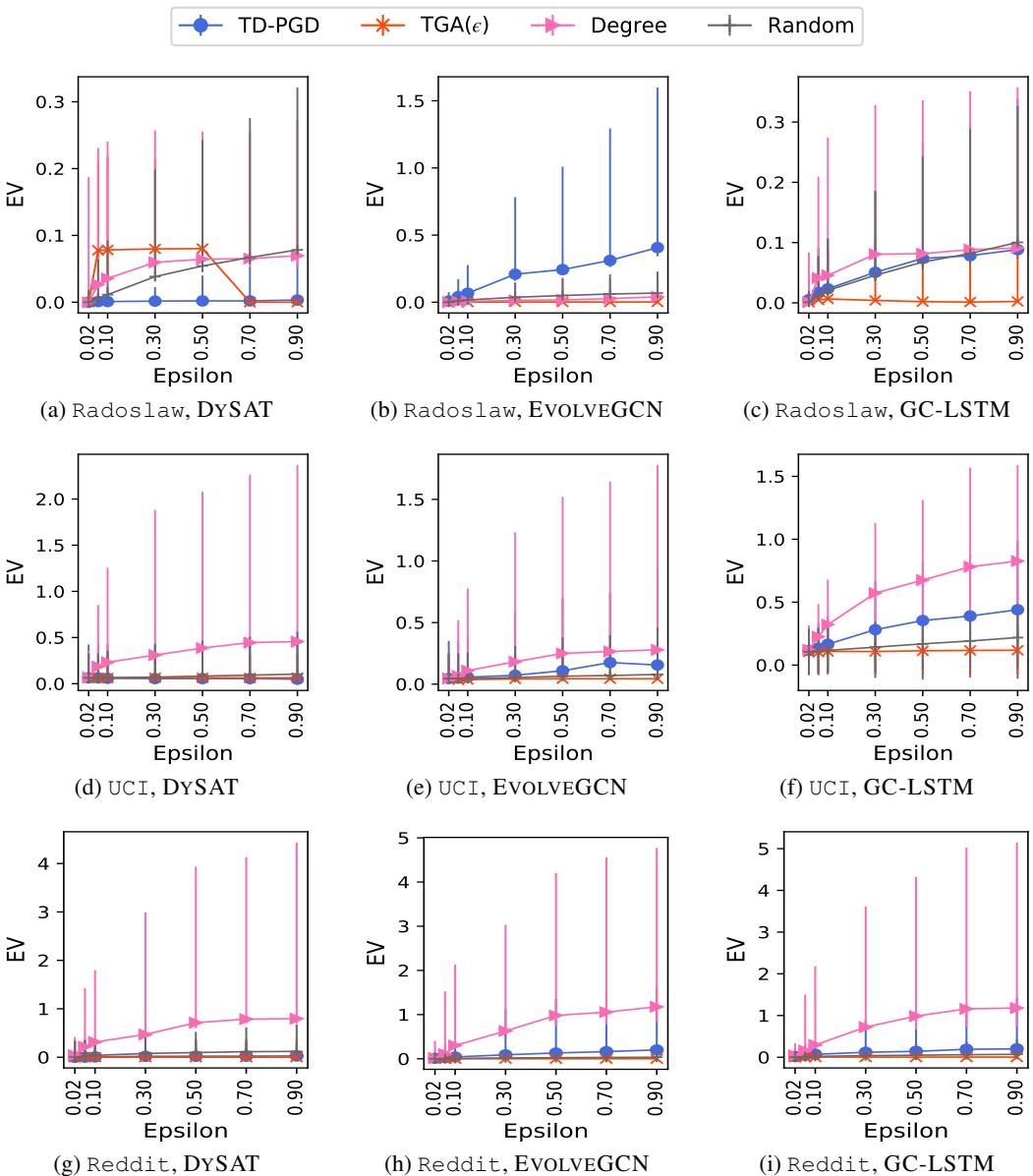

Figure 7: Comparison of embedding variability (EV) on link prediction task across datasets and models. We show median values with the interpercentile range as error bars (10% and 90%).

| Dataset | Model | Rel. Drop % | EV |
|---------|-------|-------------|-----|
| Radoslaw | DYSAT | 47.03 (5.42) | 0.00 (0.00, 0.04) |
| | EVOLVEGCN | 91.61 (3.58) | 0.25 (0.03, 0.79) |
| | GC-LSTM | 52.63 (5.09) | 0.07 (0.01, 0.19) |
| UCI | DYSAT | 4.02 (2.08) | 0.06 (0.01, 0.39) |
| | EVOLVEGCN | 96.21 (0.17) | 0.11 (0.01, 0.62) |
| | GC-LSTM | 16.12 (0.75) | 0.20 (0.03, 0.85) |
| Reddit | DYSAT | 23.24 (4.18) | 0.02 (0.00, 0.34) |
| | EVOLVEGCN | 79.31 (3.13) | 0.13 (0.02, 1.23) |
| | GC-LSTM | 15.59 (1.28) | 0.13 (0.02, 0.84) |

Table 5: Comparison of attack performance and detectability (refer Section 4) for TD-PGD at $\epsilon = 0.5$. Median values are noted with 10% and 90% quantile values in the parentheses.

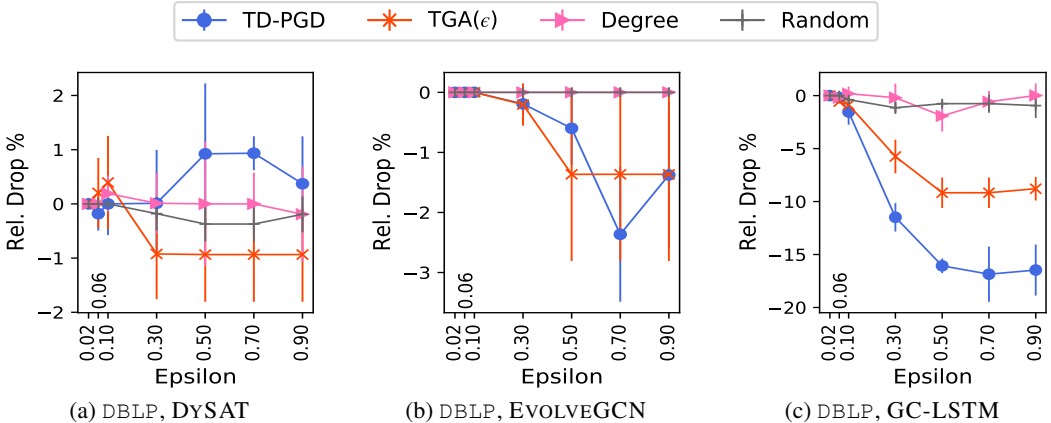

Figure 8: Structural perturbation performance on node classification task over random targets.

GC-LSTM by a factor of 2. No attack method is found to achieve a significant drop, i.e., below 5%, in the accuracy of DYSAT and EVOLVEGCN on DBLP.

**Feature perturbations**  Further analysis shows that feature perturbations are more effective in this task for random targets. Here, the task is to introduce continuous perturbations $\mathbf{S}_t^X = \boldsymbol{\mathcal{X}}'_t - \boldsymbol{\mathcal{X}}_t$ such that $\|\boldsymbol{\mathcal{X}}'_t - \boldsymbol{\mathcal{X}}_t\| \leq \epsilon d \boldsymbol{\mathcal{X}}_t$ for all $t$. Equation 1, thus, becomes

$$\max_{\boldsymbol{\mathcal{X}}'_1, \boldsymbol{\mathcal{X}}'_2, \cdots, \boldsymbol{\mathcal{X}}'_{T-1}} \mathcal{L}_{\text{task}}\left(\widehat{y}_{\text{task}}(f_{\mathcal{M}}(\boldsymbol{\mathcal{X}}'_{1:T-1})), y_{\text{task}}, E_{tg}\right) \tag{10}$$

$$\text{such that } \forall t \in (1, T) : \frac{\|\boldsymbol{\mathcal{X}}'_t - \boldsymbol{\mathcal{X}}_t\|}{\|\boldsymbol{\mathcal{X}}_t - \boldsymbol{\mathcal{X}}_{t-1}\|} \leq \epsilon$$

$$\|\boldsymbol{\mathcal{X}}'_1 - \boldsymbol{\mathcal{X}}_1\| \leq \varepsilon_1.$$

We adapt the Algorithm 1 to this problem to find effective perturbations. In particular, we replace $\mathbf{s}_t$ to denote the feature perturbation vector at time $t$, i.e., the vector corresponding to $\boldsymbol{\mathcal{X}}'_t - \boldsymbol{\mathcal{X}}_t$. Finally, since the perturbations are supposed to be in continuous space, we remove the rounding step (line 4) and return the matrix form of $\{\mathbf{s}_t\}$ as $\mathbf{S}^X$. Thus, TD-PGD can be used to find effective attacks in the feature perturbation setting as well.

However, since TGA($\epsilon$) and DEGREE takes decisions based on the structure, we omit these baselines for this setting. We use RANDOM to introduce uniformly random perturbations in the feature matrices of the random nodes.

Figure 9 compares the attack performance of the two feature perturbation methods on DBLP-5 over random targets. One can note that TD-PGD is able to achieve around 30% drop for all the models

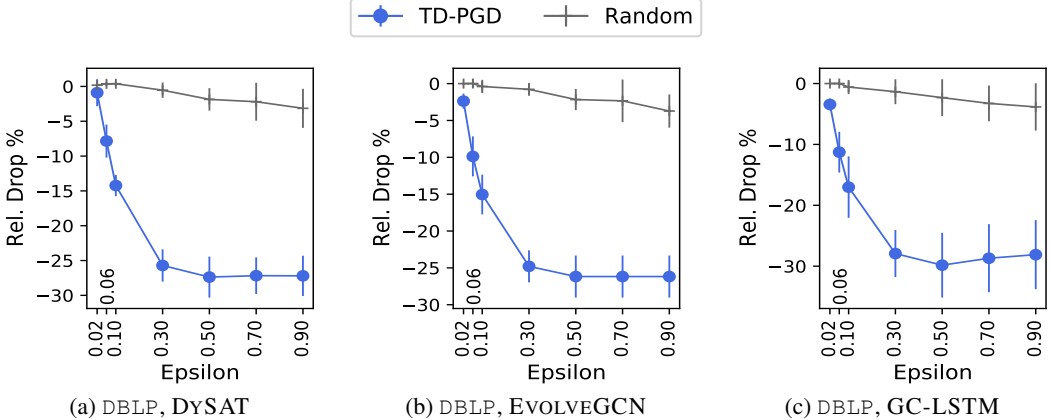

Figure 9: Feature perturbation performance on node classification task over random targets.

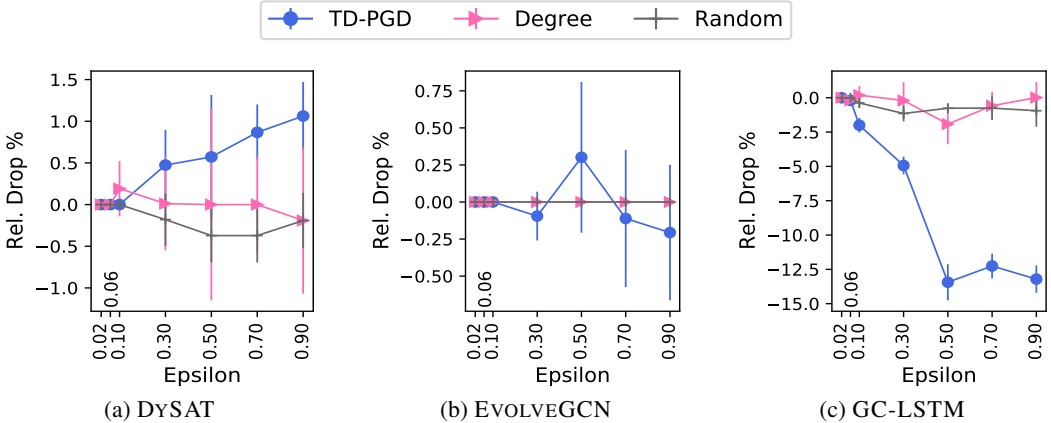

Figure 10: Online adversarial attacks on random targets for DBLP.

while it could not drop the performance below 5% for these random targets using structural perturbations. This can be explained by the low degree of these targets (average $\sim 10$ over 10 time steps) which allows for a small no. of perturbations per time step according to the TDAP constraint. Furthermore, the node features here correspond to the word2vec attributes of the author papers while the labels represent the field of the author. Thus, there is a strong connection between the attributes and the downstream labels, which makes feature perturbation more effective than structural co-author perturbations to flip predicted labels for the classification task.

## H.5 ONLINE ADVERSARIAL ATTACKS

In this section, we show the performance of the 3 methods in the online setting for node classification on the DBLP dataset. Figures 10 and 11 show performance over random and top-degree targets respectively. One can note that TD-PGD shows competitive performance over other baselines while obtaining a 5x gain in the relative drop for GC-LSTM.

## H.6 RUNNING TIME

Figure 12 compares the running time per target for different attack methods on the largest dataset, Reddit. The times are averaged over 200 targets from 3 different seeds and error bars note the standard deviation. $\text{TGA}(\epsilon)$ is the most expensive method in terms of time and scales almost linearly with $\epsilon$ (capped at 300 s). TD-PGD takes around half the time than $\text{TGA}(\epsilon)$ and remains constant with increase in $\epsilon$. This trend can be attributed to the difference in the complexity of the two methods, as shown in Appendix G.

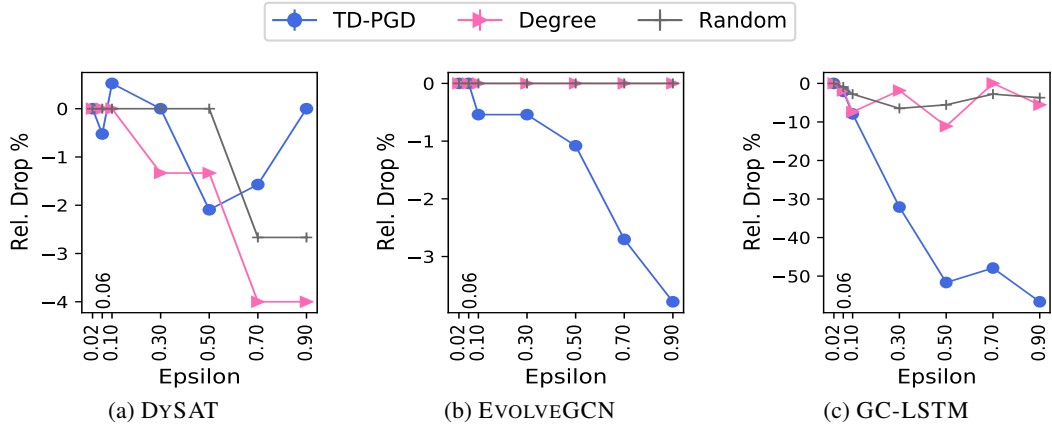

Figure 11: Online adversarial attacks on top degree targets for `DBLP`.

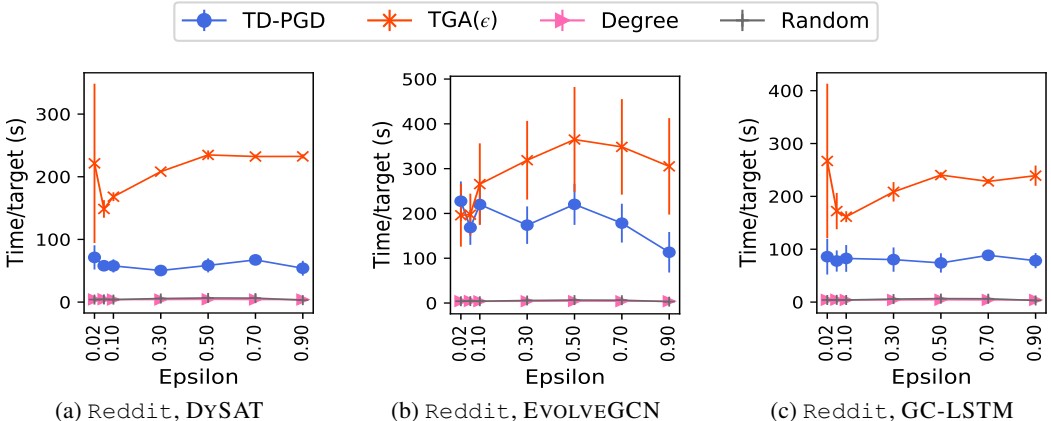

Figure 12: Running time of different attack methods on the largest dataset (`Reddit`)

Figure 13 compares the average running time of different methods at varying size of the dataset for a fixed $\epsilon$ (here = 0.5). As noted in Table 3, `Radoslaw` is the most dense and `Reddit` is the largest dataset in terms of edges. We find that TD-PGD scales at a smaller rate than TGA($\epsilon$) across different victim models.

## I  TIME-AWARE PERTURBATION: DISCUSSION

**Clarifications**: A targeted attack requires the derivative to be with respect to targets only. Thus, $\|\mathcal{A}_t - \mathcal{A}_{t-1}\|$ in Equation 2 considers adjacency matrices over the target nodes in the set $E_{tg}$ only. An untargeted attack, on the other hand, would take the differential norm over all the entities in the input (i.e., all the nodes in a graph).

## J  EXTENDED RELATED WORK

**Adversarial attacks on static graphs.**  Various attack environments, based on attacker's knowledge and intention, are studied in the literature to assess model performance under adversarial attacks on static graphs (Jin et al., 2020). In the literature, adversarial attacks are studied in different settings based on (a) attacker's knowledge of the underlying model (white-box v/s black-box), (b) their intention for the attack (targeted v/s untargeted), and (c) the timing of their attack (poisoning v/s evasion) (Jin et al., 2020). Each such combination allows us to assess the model's performance in different attack environments.

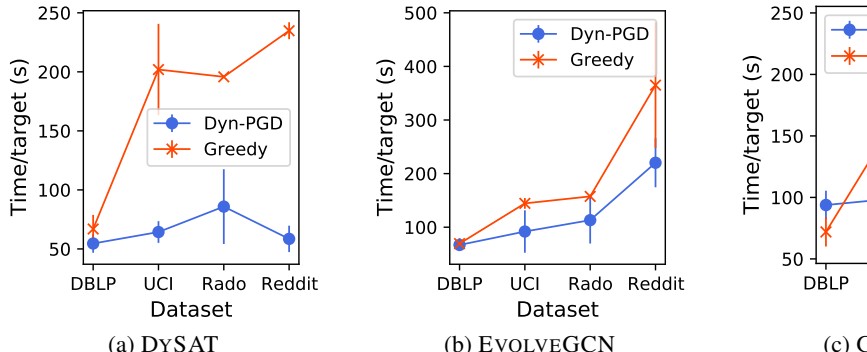

Figure 13: Running time of different attack methods at $\epsilon = 0.5$. Here, Rado means Radoslaw.

We consider a white-box, targeted, evasion setting where the objective is to introduce targeted perturbations to an already trained, fixed victim model assuming complete knowledge about its architecture and parameters. Optimization approaches for white-box evasion attacks in static graph literature include PGD (Xu et al., 2019b) and IG-JSMA (Wu et al., 2019). In the present work, we show how to use PGD to generate effective perturbations on dynamic graphs under our novel TDAP constraint. Secondly, IG-JSMA is proposed for the case of static graphs under a global budget constraint. It is not clear how the importance scores calculated for each edge in the graph sequence would be chosen in a greedy manner to hold our local TDAP constraint.

**Adversarial attacks on dynamic graphs.** Adversarial attack on dynamic graphs is an underexplored problem and no direct baselines exist that study attacks on dynamic graphs that can preserve temporal evolution (in particular, the proposed TDAP constraint). We only found three works that study adversarial attacks on dynamic graphs (Chen et al., 2021a;b; Fan et al., 2020). Out of these, only Chen et al. (2021b) considers a white-box evasion (test-time) attack, which is the same setting as ours. We adopt the greedy strategy proposed in Chen et al. (2021b) to our constraint to find feasible perturbations for our problem setting as the TGA($\epsilon$) baseline. Other methods either consider black-box attacks (Fan et al., 2020) or train-time (backdoor) attacks (Chen et al., 2021a), which cannot be directly applied here. This is because white-box attacks assume full knowledge of the victim model while black-box attacks find perturbations in a model-agnostic manner, which would be suboptimal in a setting where model knowledge is available. Secondly, backdoor attacks considers the objective of finding a trigger sequence of subgraphs that is added to the train graphs such that the model trained on these perturbed train graphs misclassifies a target link. In contrast, we introduce edge-level perturbations directly to the end-points of the target link (instead of subgraphs) during test time which implies that the trained model parameters are not updated.

**Imperceptible Attacks.** In addition to bounding the total number of perturbations with a budget, other strategies have been developed to make the attacks imperceptible for static graphs. These include rewiring the perturbed edges (Ma et al., 2019; 2021), perturbing the low degree nodes (Ma et al., 2020a), and preserving the degree/feature distribution statistics (Zügner et al., 2018).

## K ADDITIONAL PROOFS

**Theorem 4** $\overline{d\mathcal{A}'} \geq |1 - 2\epsilon|\overline{d\mathcal{A}}$

**Proof.** TDAP ensures that $\|\mathcal{A}'_t - \mathcal{A}_t\|_1 \leq \epsilon \|\mathcal{A}_t - \mathcal{A}_{t-1}\|_1$. Now, we note that since $\|\cdot\|_1 \geq \|\cdot\|_2$, we also get $\|\mathcal{A}'_t - \mathcal{A}_t\|_2 \leq \epsilon \|\mathcal{A}_t - \mathcal{A}_{t-1}\|_1$.

Further note that $\|\mathbf{x} - \mathbf{y}\|_2 \leq c$ implies $\mathbf{y} - c\mathbf{e}_r \leq \mathbf{x} \leq \mathbf{y} + c\mathbf{e}_r$ for all unit vectors $\mathbf{e}_r$ ($\|\mathbf{e}_r\| = 1$). Thus, $\|\mathcal{A}'_t - \mathcal{A}_t\|_2 \leq \epsilon \|\mathcal{A}_t - \mathcal{A}_{t-1}\|_1$ implies $\mathcal{A}_t - \epsilon\|\mathcal{A}_t - \mathcal{A}_{t-1}\|_1 \mathbf{e}_r \leq \mathcal{A}'_t \leq \mathcal{A}_t + \epsilon\|\mathcal{A}_t - \mathcal{A}_{t-1}\|_1 \mathbf{e}_r$.

Substituting the above inequalities for $\mathcal{A}'_t$ and $\mathcal{A}'_{t-1}$ in $\|\mathcal{A}'_t - \mathcal{A}'_{t-1}\|_1$, we get $\|\mathcal{A}'_t - \mathcal{A}'_{t-1}\|_1 \geq \|\mathcal{A}_t - \mathcal{A}_{t-1} - \epsilon\|\mathcal{A}_t - \mathcal{A}_{t-1}\|\mathbf{e}_{r,1} - \epsilon\|\mathcal{A}_{t-1} - \mathcal{A}_{t-2}\|\mathbf{e}_{r,2}\|_1$, for some unit vectors $\mathbf{e}_{r,1}, \mathbf{e}_{r,2}$.

Using reverse triangle inequality, $\|\mathcal{A}'_t - \mathcal{A}'_{t-1}\|_1 \geq \left| \|\mathcal{A}_t - \mathcal{A}_{t-1}\|_1 - \epsilon\|\|\mathcal{A}_t - \mathcal{A}_{t-1}\|_1 \mathbf{e}_{r,1} + \right.$

$\|\mathcal{A}_{t-1} - \mathcal{A}_{t-2}\|_1 \mathbf{e}_{r,2}\|_1\Big| \;\geq\; \Big|\|\mathcal{A}_t - \mathcal{A}_{t-1}\|_1 - \epsilon(\|\mathcal{A}_t - \mathcal{A}_{t-1}\|_1\|\mathbf{e}_{r,1}\|_1 + \|\mathcal{A}_{t-1} - \mathcal{A}_{t-2}\|_1\|\mathbf{e}_{r,2}\|_1)\Big|$, where the last inequality is by triangle inequality. Summing both sides over all time steps, we get $\sum_t\|\mathcal{A}'_t - \mathcal{A}'_{t-1}\|_1 \;\geq\; \sum_t\Big|\|\mathcal{A}_t - \mathcal{A}_{t-1}\|_1 - \epsilon(\|\mathcal{A}_t - \mathcal{A}_{t-1}\|_1 + \|\mathcal{A}_{t-1} - \mathcal{A}_{t-2}\|_1)\Big| \;\geq\; \Big|\sum_t\|\mathcal{A}_t - \mathcal{A}_{t-1}\|_1 - \epsilon(\|\mathcal{A}_t - \mathcal{A}_{t-1}\|_1 + \|\mathcal{A}_{t-1} - \mathcal{A}_{t-2}\|_1)\Big|$. Replacing $\sum_t\|\mathcal{A}_t - \mathcal{A}_{t-1}\|$ as $\overline{d\mathcal{A}}$, we get $\overline{d\mathcal{A}'} \geq |1 - 2\epsilon|\overline{d\mathcal{A}}$.

∎

**Theorem 5** $\|\mathbf{Z}'_t - \mathbf{Z}_t\| \leq C\epsilon\|\mathcal{A}_t - \mathcal{A}_{t-1}\|$

**Proof.** By Cauchy's MVT, $\mathbf{Z}'_t - \mathbf{Z}_t \leq \nabla f \cdot \mathbf{q}'_t - \mathbf{q}_t$, which gives us $\|\mathbf{Z}'_t - \mathbf{Z}_t\| \leq \|\nabla f\|\,\|\mathbf{q}'_t - \mathbf{q}_t\|$ by Cauchy-Schwarz inequality. Note that $\|\mathbf{q}'_t - \mathbf{q}_t\|_1 = \|\mathcal{A}'_t - \mathcal{A}_t\|_1$. Thus, $\|\mathbf{Z}'_t - \mathbf{Z}_t\| \leq \|\nabla f\|\,\|\mathcal{A}'_t - \mathcal{A}_t\| = C\|\mathcal{A}'_t - \mathcal{A}_t\| \leq C\epsilon\|\mathcal{A}_t - \mathcal{A}_{t-1}\|$, by the definition of TDAP constraint. ∎

**Theorem 6** $d\mathbf{Z}' \geq \chi d\mathbf{Z}$

**Proof.** Note that $\mathbf{Z}_t = f(\mathcal{A}_t, \mathcal{A}_{t-1}, \cdots, \mathcal{A}_1)$. We consider a stacked vector of flattened matrices $\forall \tau \in [0, t] : \mathbf{q}_{\leq\tau} = (\mathbf{q}_\tau, \mathbf{q}_{\tau-1}, \cdots, \mathbf{q}_1, \mathbf{0}, \mathbf{0}, \cdots, \mathbf{0})$, where $\mathbf{q}_i$ is the flattened vector of $\mathcal{A}_\tau$ and we append $(t - \tau)$ $\mathbf{0}$s to make all vectors $\mathbf{q}_{\leq i}$ of fixed dimension $t$. Then, by Cauchy's Mean Value Theorem in several variables, we have $\mathbf{Z}_t - \mathbf{Z}_{t-1} \geq \nabla f_{t-1} \cdot (\mathbf{q}_{\leq t} - \mathbf{q}_{\leq t-1})$, which gives us $\|\mathbf{Z}_t - \mathbf{Z}_{t-1}\| \geq \|\nabla f_{t-1}\|\,\|\mathbf{q}_{\leq t} - \mathbf{q}_{\leq t-1}\|\cos(\theta)$ by the definition as $\theta$ is the angle between $\mathbf{q}_{\leq t}$ and $\mathbf{q}_{\leq t-1}$. We note that $\|\mathbf{q}_{\leq t} - \mathbf{q}_{\leq t-1}\|_1 = \|(\mathbf{q}_t - \mathbf{q}_{t-1}, \cdots, \mathbf{q}_2 - \mathbf{q}_1, \mathbf{q}_1)\|_1 = \sum_t\|\mathbf{q}_t - \mathbf{q}_{t-1}\|_1 = T\overline{d\mathcal{A}}$. Thus, we have $\|\mathbf{Z}_t - \mathbf{Z}_{t-1}\|_1 \geq C_3\overline{d\mathcal{A}}$ for some constant $C_3 \geq 0$. Using Theorem 4, we get $d\mathbf{Z}' \geq C_3\overline{d\mathcal{A}'} \geq C_3|1 - 2\epsilon|\overline{d\mathcal{A}}$. By mean-value theorem and Cauchy-Schwarz inequality, we also have $d\mathbf{Z}_t \leq \|\nabla f\|\,\|\mathbf{q}_{\leq t} - \mathbf{q}_{\leq t-1}\| = C_4\overline{d\mathcal{A}}$. Hence, $d\mathbf{Z}' \geq C_3|1 - 2\epsilon|\overline{d\mathcal{A}} \geq \frac{C_3}{C_4}|1 - 2\epsilon|d\mathbf{Z} = \chi d\mathbf{Z}$ for some constant $\chi \geq 0$. ∎

## L   Anomaly Detection With TDAP Constraint

Anomaly detection methods for dynamic graphs can be divided into two broad categories based on the available input Akoglu et al. (2015); Ma et al. (2021); Ranshous et al. (2015).

**Supervised methods.** Ground-truth labels for anomalous edges are known or estimated from the data. A model is trained to minimize a supervised loss to classify the labeled edges in the training set. For example, Cai et al. (2021); Zhu et al. (2020) use Cross Entropy loss while Zheng et al. (2019); Wang et al. (2020) use margin loss on a parameterized anomalous score that is calculated for each edge using end-to-end trained node embeddings. Since these anomaly detection methods are trained on supervisory signals that may not be accessible to an attacker, evading these anomaly detectors is not the focus of our work.

**Unsupervised methods.** Here, anomalies are detected by studying certain properties of the dynamic graph in the structural or embedding space. Victim graph representation models can employ such unsupervised strategies as a defense mechanism against adversarial perturbations. Thus, an attacker must be able to defend against such methods to be effective in practice. The focus of our current work is thus to make the unsupervised methods fail to detect adversarial edges as anomalies.

Traditional methods have studied, different graph distance metrics between consecutive snapshots such as graph edit distance, hamming distance, and spectral distance, to detect anomalous snapshots Shoubridge et al. (2002); Bunke et al. (2007). If the distance with the previous snapshot exceeds a threshold then the instance is deemed anomalous. These papers, however, have focused on effectively detecting graph anomalies instead of edge anomalies at any point in time. In our work, we consider a targeted attack setting, where perturbations (anomalies) are added to only change the local behavior around the target and not the global graph dynamics. A graph-level anomaly detection would thus not be able to detect local perturbations. For this reason, we study these distance metrics for just the ego-network around the target node. Theorems 1 and 4 show that the average rate of structural change remains preserved within certain factors that can be tuned to fool these detectors. Suppose one had chosen a threshold $B$ such that $d\mathbf{A}_t \geq B$ is defined an anomaly. If initially the graph sequence was not anomalous on average, i.e., $\overline{d\mathbf{A}} \leq B$, then after perturbation, $\overline{d\mathbf{A}'} \geq B$ would

happen if $\overline{d\mathbf{A}'} \geq |1 - 2\epsilon|\overline{d\mathbf{A}} \geq B$, i.e., $\overline{d\mathbf{A}} \geq B/(|1 - 2\epsilon|)$. Thus, one can choose a value of $\epsilon$ such that this does not hold for a given threshold $B$ and snapshot dynamics $\overline{d\mathbf{A}}$.

Embedding-based anomaly detection strategies have also been proposed that work in an unsupervised manner. Yu et al. (2018) clusters the embeddings of a sampled edge set at each timestep and flags an edge to be anomalous if its distance in the embedding space from each cluster exceeds a threshold. One can note that TDAP-constrained solutions would be effective against such a detector as well since as noted in Theorem 5, embeddings before and after perturbation change only by a small factor of the actual change in the adjacency matrices relative to the previous timestep. Thus, the distance of edge embeddings from the cluster centers would change according to the structural evolution at that time. We further show this with empirical evidence as below.

Another embedding-based method uses the norm distance between embeddings in consecutive snapshots to find anomalies when the distance exceeds a threshold Goyal et al. (2018). Theorems 2 and 6 directly note that the embedding distance between consecutive snapshots would be preserved. We also show empirical evidence of this preservation via our metric Embedding Variability. Note that Embedding Variability measures the ratio of the range of the consecutive embedding distance before and after the perturbation. Thus, if the embedding variability is $\kappa$, it means the range of $d\mathbf{Z}_\tau$ after perturbation is $1 + \kappa$ (or $1 - \kappa$) times the range of $d\mathbf{Z}_\tau$ before perturbation. If the threshold was set to be $D$, then our perturbation would be anomalous if $D \geq d\mathbf{Z}_\tau \geq D/(1 + \kappa)$. Thus, we can choose an $\epsilon$ such that the obtained EV (i.e., $\kappa$) does not satisfy the above inequality for a given threshold $D$ and embedding evolution $d\mathbf{Z}_\tau$. In this work, we show how $\kappa$ varies with different $\epsilon$ and can be as low as 0.04 even for an $\epsilon = 0.5$.

### L.1 Empirical Analysis

**Netwalk.** In this section, we test the efficacy of Netwalk anomalous scores to detect perturbations made under the TDAP constraint. We used the 3 victim representation models (as above) to obtain the embeddings and used the K-Means algorithm with $k = 5$ for clustering at each time step. Perturbations are selected from the TD-PGD algorithm and the edge embeddings are clustered in the original and perturbed embedding space for a fixed set of held-out training edges. The anomalous score is then calculated as the average distance of the perturbed edges to the nearest cluster's centroid in the corresponding embedding space.

Figure 14 shows anomalous score of the Netwalk algorithm Yu et al. (2018) of the perturbed edges in different victim models for different datasets. One can note that the anomalous score in the perturbed embedding space almost perfectly overlaps the anomalous score in the original embedding space. One exception is Figure 14b where the distance after perturbation increases significantly over the original space as $\epsilon$ increases. We also conducted a 2-sample t-test between original and perturbed anomalous scores at each epsilon value for each model and dataset. The hypothesis of the two distributions being similar was accepted (i.e., $p$-value $> 0.05$) in all but five cases. These were EvolveGCN for Radoslaw at $\epsilon = 0.3, 0.5, 0.7, 0.9$ and DySAT for Radoslaw at $\epsilon = 0.9$. Thus, perturbations under the TDAP constraint are not easily detectable by the Netwalk algorithm.

**DynGem.** DynGem detects anomalies based on the value $d\mathbf{Z}_t$ at each time $t$. While Embedding Variability gives an idea of how different the distribution of $d\mathbf{Z}_t$ is before and after perturbation, the value can still be arbitrarily low. Thus, we conduct a 2-sample t-test between $d\mathbf{Z}$ and $d\mathbf{Z}'$ for each model-dataset pair at different epsilons, to see if the distributions are different. Table 6 show $p$-values for each case and highlights the cases where the null hypothesis of the two distributions was not rejected (i.e., $p$-value $> 0.05$). One can note that the t-test found a difference in $d\mathbf{Z}$ in only 40 out of 117 cases across 3 datasets, 3 victim models, and 13 epsilon values.

## M Randomized Rounding for TD-PGD

We use randomized rounding for TD-PGD in order to efficiently obtain a valid discrete perturbation solution from the continuous vector obtained by running $N$ iterations of the PGD loop. Inspired by existing works on using PGD for graphs Xu et al. (2019b); Geisler et al. (2021), we do randomized rounding for a fixed number of iterations and pick the solution that maximizes the loss while satisfying the constraint. Furthermore, to ensure fast convergence, we adopt the top-k heuristic sampling strategy in the first iteration Geisler et al. (2021). Algorithm 4 describes the steps taken in TD-PGD

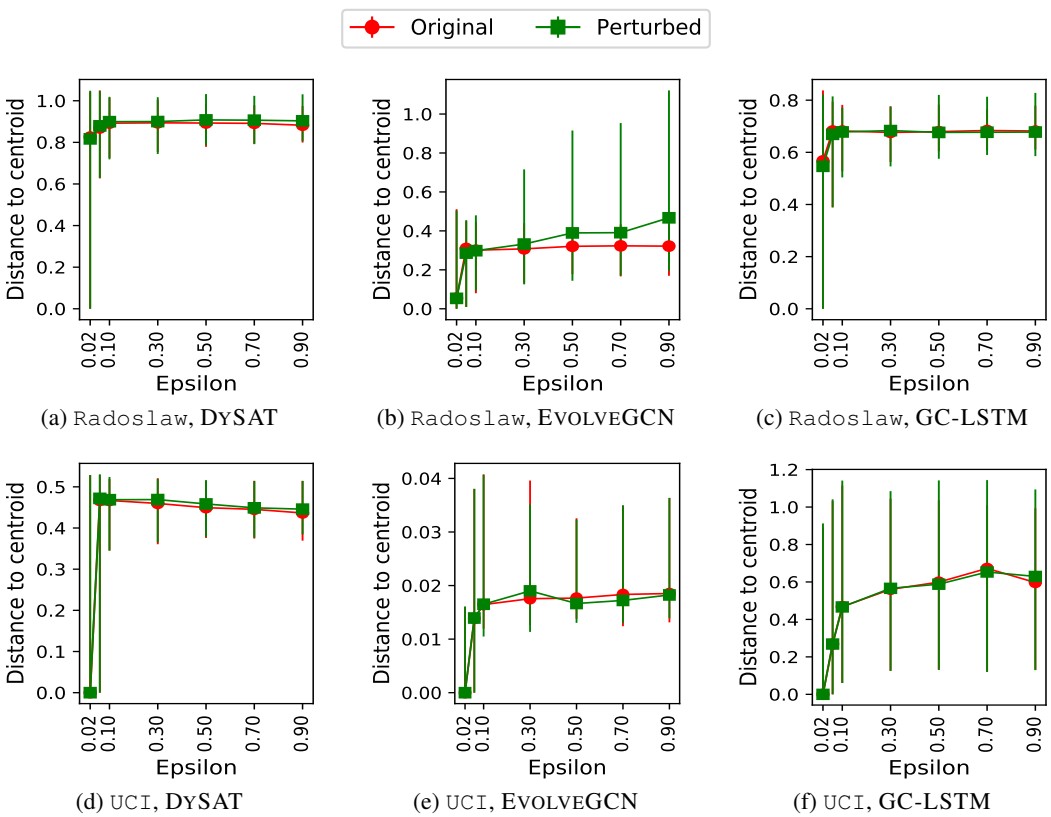

Figure 14: Distance of the embeddings to the nearest centroid for the perturbed edges at different epsilons before and after perturbation.

| Dataset | Model | Epsilon | $p$-value $(d\mathbf{Z}', d\mathbf{Z})$ |
|---|---|---|---|
| | **DYSAT** | **[0.02, 0.9]** | **≥ 0.8387** |
| | **EVOLVEGCN** | **[0.02, 0.2)** | **≥ 0.2054** |
| Radoslaw | EVOLVEGCN | [0.2, 0.9] | ≤ 0.0002* |
| | **GC-LSTM** | **[0.02,0.3)** | **≥ 0.1704** |
| | GC-LSTM | [0.3, 0.9] | ≤ 0.0432* |
| | **DYSAT** | **[0.02, 0.9]** | **≥ 0.9252** |
| | **EVOLVEGCN** | **[0.02, 0.3)** | **≥ 0.0878** |
| UCI | EVOLVEGCN | [0.3, 0.9] | ≤ 0.0197* |
| | **GC-LSTM** | **[0.02, 0.2)** | **≥ 0.2674** |
| | GC-LSTM | [0.2, 0.9] | ≤ 0.0378* |
| | **DYSAT** | **[0.02, 0.9]** | **≥ 0.9636** |
| | **EVOLVEGCN** | **[0.02, 0.3)** | **≥ 0.0735** |
| Reddit | EVOLVEGCN | [0.3, 0.9] | ≤ 0.0271* |
| | **GC-LSTM** | **[0.02, 0.6)** | **≥ 0.0554** |
| | GC-LSTM | [0.6, 0.9] | ≤ 0.0270* |

Table 6: Significance values from 2-sample t-test between $d\mathbf{Z}'$ and $d\mathbf{Z}$. Bold rows indicate that the difference is not significant and thus, DynGem-based anomalous scoring would be ineffective to detect TDAP perturbations as made by our method TD-PGD.

to obtain a valid discrete solution. Note that in the case of online TD-PGD, this algorithm runs for a single time step $t$ instead of $T - 1$.

---

**Algorithm 4** ROUND($\mathbf{s}, N_r, \{\varepsilon_t\}$)

---

**Require:** Perturbation vector $\mathbf{s}$, Number of randomized iterations $N_r$, TDAP variables $\varepsilon_t$.

1:  $\mathbf{S}_t \leftarrow \mathbf{0}$ for all $t$
2:  $\mathcal{L}_{best} \leftarrow -\infty$
3:  **for** $t = T - 1$ to 1 **do**
4:      $P_t \leftarrow$ Select the top $\lfloor \varepsilon_t \rfloor$ perturbations based on the values of $\mathbf{s}_t$.
5:      $\mathbf{S}_t[k] = 1$ if $k \in P_t$ otherwise 0.
6:      $\mathcal{L}_{best} \leftarrow \mathcal{L}_{task}(\{\mathcal{G}_t \oplus \mathbf{S}_t; \ \forall t\}, y_{task}, E_{tg})$
7:  **for** $i = 2$ to $N_r$ **do**
8:      $\mathbf{S}_t[i] \sim$ Bernoulli($\mathbf{s}_t$) for each $t$
9:      **if** $\|\mathbf{S}_t[i]\| \le \varepsilon_t$ for all $t$ **then**
10:         $\mathcal{L}[i] \leftarrow \mathcal{L}_{task}(\{\mathcal{G}_t \oplus \mathbf{S}_t[i]; \ \forall t\}, y_{task}, E_{tg})$
11:         **if** $\mathcal{L}[i] > \mathcal{L}_{best}$ **then**
12:             $\mathbf{S}_t \leftarrow \mathbf{S}_t[i], \mathcal{L}_{best} \leftarrow \mathcal{L}[i]$
13: **return** $\mathbf{S}_t$ for all $t$.

---

# N    POTENTIAL NEGATIVE SOCIETAL IMPACTS

Our work presents the first comprehensive study on adversarial attacks on discrete-time dynamic graphs and shows that existing learning models in this domain are vulnerable to even those attacks that can preserve original evolution of graphs. Dynamic graphs can model a large variety of socially critical data structures, ranging from social media, epidemiology, finance, biology, and road networks. Our work shows that state-of-the-art deep learning prediction models are vulnerable to perturbations under the TDAP constraint. This means that these models are not suitable for deployment in environments where adversarial attacks under a TDAP constraint are realistic. If these models are deployed in such a setting, adversarial attacks proposed in this work can be used by an adversary to hamper the predictions. However, it must be noted that none of the models used in this work are known to be deployed in the real-world. Secondly, TDAP constraint with a large $\epsilon$ may not be realistic in many applications. Thus, our study has an overall positive impact on the society as it allows practitioners to test the robustness of their models under a practical constraint before deploying in vulnerable environments. Although our attacks can be adopted by adversaries for negative use, it is important to put it out in the community so that the model designers are made aware of these attacks that are specifically designed to evade detection.

