# OpenReview forum: "Temporal Dynamics Aware Adversarial Attacks On Discrete-Time Graph Models"
_ICLR.cc/2023/Conference — Submitted to ICLR 2023_

### Official Review · Reviewer_pnFA · 2022-10-23

**Confidence:** 2
**Correctness:** 4
**Technical Novelty And Significance:** 2
**Empirical Novelty And Significance:** 2
**Recommendation:** 1

**Clarity, Quality, Novelty And Reproducibility:**

* Low quality, low novelty.
* Clarity and reproducibility are fine.


**Strength And Weaknesses:**

Strengths


Weaknesses

* The paper and, in particular, the numerical studies do not provide much motivation as to why we need to study adversarial attacks in this setting.
* The theory. Proposition 1 does not appear to be a true proposition but a condition. Are the Theorems really important? How good is the PGD approach at soolving the optimization problem in equation (2) in any specific setting?

**Summary Of The Paper:**

This paper studies adversarial attacks on time-evolving graphs. The paper introduces the Temporal Dynamics-Aware Perturbation (TDAP) constraint, which allows for small perturbations of graphs analogous with other adversarial settings, while still preserving the graph evolution. This constraint appears to be combinatorial, and the paper suggests greedy and projected gradient descent with rounding approaches. Numerical studies are shown to show the effect of the perturbations.


**Summary Of The Review:**

* This seems to be an extension of adversarial perturbations to a setting where it is unclear the study is necessary.
* Even if the problem were important, the theory seems weak and should be revised in its content and/or presentation.
* Strangely, a new proposal for a defense mechanism is missing.

---

> ### Author Response · Authors · 2022-11-16
> **Very Important and Novel Problem Setting**
>
> We very politely disagree with the reviewer’s assessment on the significance of the problem setting in our paper and the weaknesses of our contributions. For evidence, we refer the reviewer to all the 4 other reviewer assessments of this paper which highlight the novelty of the problem and solution approach in the paper, the soundness of our theoretical contributions, and strong empirical support. To address the reviewer’s generic concerns, we provide a comprehensive but succinct view of the problem setting and highlight our contributions in the specific responses below. We discuss the importance and novelty of research direction and our contributions that take a significant step on this research topic with the potential of opening new avenues. We further discuss the importance and correctness of theoretical developments in the paper. We hope that these responses are able to convince the reviewer about the significance of our contribution and request them to revisit their score in light of our responses and the other reviewers’ comments. We invite the reviewer to raise specific technical questions if any of their concerns still remain unresolved that prevent them from increasing their score and we will be happy to incorporate their feedback in the subsequent iteration.
>
> > The paper and, in particular, the numerical studies do not provide much motivation as to why we need to study adversarial attacks in this setting.
>
> > This seems to be an extension of adversarial perturbations to a setting where it is unclear the study is necessary.
>
> **Motivation of our work:** Real-world graphs are dynamic as entities interact with each other in real time [1,2] (reference number based on the list below). Dynamic graphs can represent data from multiple domains [3] such as protein-protein interactions [4], neural activation patterns [5], online social interactions [6], and e-commerce [7]. With the advent of deep learning, many dynamic graph models [8] have been proposed and are deployed for various industrial applications including the recommendation systems on Twitter [9] and Snapchat [10]. There have been over 160 papers that study adversarial attacks on static graph neural networks [11] and are published in top conference venues including ICLR, ICML, NeurIPS, and KDD. In comparison, despite the importance of dynamic graphs and the prevalence of dynamic graph neural network models, only one paper exists in the literature that studies the vulnerability of dynamic graph neural networks [12].
>
> Thus, our work bridges this gap by studying, for the first time, the robustness of state-of-the-art dynamic graph models under small undetectable perturbations. Attackers have the incentive to attack such that they are not detected for malicious purposes [13-15]. Our work formulates a novel setting that studies adversarial attacks for dynamic graphs while theoretically preserving the temporal evolution of the graphs (please refer to Theorems 1,2, 4-6). We further show that perturbations introduced under this novel constraint evade detection by state-of-the-art anomaly detection algorithms (please refer to Tables 2, 5, 6 and Figures 6, 7, 14). Thus, we believe that **our work provides strong motivation** to study temporal dynamics-aware adversarial attacks on dynamic graph models.
>
> **Contributions of our work:** In terms of our contributions, we refer the reviewer to Table 1 of our paper and note that this is the first work to study **targeted, white-box, evasion** attacks on **dynamic** graph models. In this paper, we make significant advancements in this novel setting on  three fronts. (1) Ours is the first work that studies temporal-dynamics aware adversarial attacks for dynamic graphs, i.e., attacks that can theoretically preserve graph evolution. (2) We are the first to study online adversarial attacks for dynamic graph models, a highly practical but challenging setting since the attacker perturbs the graphs in real time without having access to future snapshots and without changing the past snapshots. (3) We also empirically show the effectiveness of our temporal dynamics-aware attacks against being undetected by anomaly detection algorithms. Such an analysis has never been conducted for adversarial attacks on dynamic graphs and we believe that our contributions will help the community study the robustness of state-of-the-art dynamic graph representation models in practical settings. We, therefore, believe that this is the first principled and state-of-art approach for an important and challenging research topic of attack on dynamic graphs, that is still in its infancy and has great potential avenues for further research.

---

> > ### Author Response · Authors · 2022-11-16
> > **Substantial Theoretical and Empirical Contributions**
> >
> > > The theory. Proposition 1 does not appear to be a true proposition but a condition. Are the Theorems really important? How good is the PGD approach at soolving the optimization problem in equation (2) in any specific setting?
> >
> > > Even if the problem were important, the theory seems weak and should be revised in its content and/or presentation.
> >
> > Proposition 1 states that to preserve graph evolution through perturbations, the number of perturbations must be no more than $\epsilon$ times the differential at time $t$. Through this, we propose a novel constraint on perturbations $\mathbf{A}’_t$ such that $\lVert \mathbf{A}’_t - \mathbf{A}_t \rVert \le \epsilon d \mathbf{A}_t$ must happen for all $t$ to preserve temporal evolution, which we prove is true from Theorems 1, 2, 4-6. These theorems form the backbone of our main claims and are crucial to establish the validity of our proposed constraint for our objective discussed in Section 3. In particular, theorems validate the claims that TDAP constraint ensures that the introduced perturbations do not disrupt the evolving trend of dynamic graphs both in the graph structure presented by adjacency matrix and graph representations learned through victim models.
> >
> > We show that the PGD approach is good at solving the optimization problem of Equation 2 through experiments on 4 datasets across 2 tasks. Please refer to Figure 1 for results on dynamic link prediction and Figure 2 for results on node classification results. These show an improvement of up to $4$x over existing baselines. We also show the effectiveness of the PGD algorithm in a more practical and challenging online setting where perturbations are made in real time without knowledge of future snapshots and without changing the past snapshots. Figure 3 shows superior performance of PGD over representative baselines (with up to $5$x improvement).
> >
> > In general, our theoretical proofs have been seen as a strength by other reviewers (please see reviewers kVti, u4F7), which we have further strengthened even more based on their feedback. If you could provide any specific weaknesses in the theory, we would be happy to revise and iterate with you.
> >
> > > Strangely, a new proposal for a defense mechanism is missing.
> >
> > Most adversarial attack papers (over 150 out of 160) published at top venues such as ICLR, NeurIPS, KDD, AAAI, ICML, etc. solely focus on studying adversarial attacks for graphs without proposing a new defense mechanism [11]. In this work, we do the first comprehensive attack for dynamic graphs and show vulnerability of state-of-the-art dynamic graph models for both offline and online settings while evading anomaly detection algorithms by preserving the temporal evolution. As we discussed in response to the first question, our paper has many important contributions that we believe will help the community study the robustness of state-of-the-art dynamic graph representation models in more practical settings. Our focus has, therefore, been on building an attack approach with rigorous theoretical foundations backed by strong empirical support. We also believe that proposing a defense mechanism as a part of this paper would fail to do justice to this new and unexplored problem setting. Following a plethora of research papers on adversarial attacks, we argue that defense mechanism is not a requirement for attack papers and are approaching it in an ongoing work that is best left as a separate paper to do justice to both concepts separately.

---

> > > ### Author Response · Authors · 2022-11-16
> > > **References for the above responses**
> > >
> > > [1] Leskovec, Jure, Jon Kleinberg, and Christos Faloutsos. "Graph evolution: Densification and shrinking diameters." ACM transactions on Knowledge Discovery from Data (TKDD) 1.1 (2007): 2-es.
> > >
> > > [2] Kossinets, Gueorgi, and Duncan J. Watts. "Empirical analysis of an evolving social network." science 311.5757 (2006): 88-90.
> > >
> > > [3] Holme, Petter, and Jari Saramäki. "Temporal networks." Physics reports 519.3 (2012): 97-125.
> > >
> > > [4] Ou-Yang, Le, et al. "Detecting temporal protein complexes from dynamic protein-protein interaction networks." BMC bioinformatics 15.1 (2014): 1-14.
> > >
> > > [5] Hutchison, R. Matthew, et al. "Dynamic functional connectivity: promise, issues, and interpretations." Neuroimage 80 (2013): 360-378.
> > >
> > > [6] Christakis, Nicholas A., and James H. Fowler. "Social contagion theory: examining dynamic social networks and human behavior." Statistics in medicine 32.4 (2013): 556-577.
> > >
> > > [7] Leskovec, Jure, Lada A. Adamic, and Bernardo A. Huberman. "The dynamics of viral marketing." ACM Transactions on the Web (TWEB) 1.1 (2007): 5-es.
> > >
> > > [8] Kazemi, Seyed Mehran, et al. "Representation learning for dynamic graphs: A survey." J. Mach. Learn. Res. 21.70 (2020): 1-73.
> > >
> > > [9] Tang, Xianfeng, et al. "Knowing your fate: Friendship, action and temporal explanations for user engagement prediction on social apps." Proceedings of the 26th ACM SIGKDD international conference on knowledge discovery & data mining. 2020.
> > >
> > > [10] Rossi, Emanuele, et al. "Temporal graph networks for deep learning on dynamic graphs." arXiv preprint arXiv:2006.10637 (2020).
> > >
> > > [11] [https://github.com/safe-graph/graph-adversarial-learning-literature](https://github.com/safe-graph/graph-adversarial-learning-literature)
> > >
> > > [12] Chen, Jinyin, et al. "Time-aware gradient attack on dynamic network link prediction." IEEE Transactions on Knowledge and Data Engineering (2021).
> > >
> > > [13] Apruzzese, Giovanni, Michele Colajanni, and Mirco Marchetti. "Evaluating the effectiveness of adversarial attacks against botnet detectors." 2019 IEEE 18th International Symposium on Network Computing and Applications (NCA). IEEE, 2019.
> > >
> > > [14] Ilyas, Andrew, et al. "Black-box adversarial attacks with limited queries and information." International Conference on Machine Learning. PMLR, 2018.
> > >
> > > [15] Apruzzese, Giovanni, et al. "Modeling realistic adversarial attacks against network intrusion detection systems." Digital Threats: Research and Practice (DTRAP) 3.3 (2022): 1-19.

---

> > > ### Comment · Reviewer_pnFA · 2022-11-17
> > > **Still disagree**
> > >
> > > Let’s start on the practical side. I remain unconvinced that this is an important problem. Additional citations won’t change that. Extending a static problem to a time-evolving problem that, in broad strokes, appears quite similar does not seem particularly novel.
> > >
> > > Onto the theory.
> > >
> > > Proposition 1 states:
> > >
> > > “The number of perturbations introduced to input x at time step t must not be more than a fraction epsilon times the differential at t,i.e TDAP :=… for all t.”
> > >
> > > Whether you intended this to be a proposition or not or meant to include an if statement along the lines of “if graph evolution is preserved through perturbations [in some precise manner]”, this is not included in the submitted paper. Very generously, it’s poor writing and must be addressed through revision and resubmission.
> > >
> > > That other reviewers see the theory as a strength is of no importance to me. I’ve seen and written enough papers with good, substantive theory to know that a few paragraphs of proofs are not a compelling reason for acceptance. In the absence of some other substantial insight. For instance, use some approximation theoretic results on rounding solutions to continuous relaxations of discrete optimization problems to quantify the suboptimality of your PGD algorithm and I’ll accept that paper. Provide a provable defense bound (with of course some loss depending on the perturbation strength) and I’ll accept that paper. Find a better attack method than PGD (which is fairly standard), and that may also be pretty interesting.

---

### Official Review · Reviewer_u4F7 · 2022-10-24

**Confidence:** 3
**Clarity, Quality, Novelty And Reproducibility:** Writing can be improved in some parts…
**Correctness:** 3
**Technical Novelty And Significance:** 3
**Empirical Novelty And Significance:** 3
**Recommendation:** 5

**Strength And Weaknesses:**

\+ It is a less studied problem that generates attacks on the time-evolving graphs.

\+ Theoretical proofs and extensive experiments are given.

\- The layout of this article is somewhat hard to follow.

**Summary Of The Paper:**

The paper aims at modeling adversarial attacks which can perturb the input graph sequence on discrete-time dynamic graph models meanwhile preserving the temporal dynamics of the graph.Then they propose a constraint named Temporal Dynamics-Aware Perturbation (TDAP) to make imperceptible perturbations in discrete-time. Through the proof provided, the perturbations under TDAP constraint preserve changes from the structural and embedding perspectives.

**Summary Of The Review:**

1. The meaning of proposed metric embedding variability (EV) is similar to the TDAP constraint. So the value of EV should be small intuitively under TDAP constraint. It is better to provide the explanation and comparison to other existed metrics.

2. Table 2 only reports the EV metric for TD-PGD, I would like to see more attack methods in the results. And the proposed method seems cannot achieve the best results in appendix H.4, whiy.

3. Some baselines should be considered, e.g., TGA (Chen et al., 2021b).

---

> ### Author Response · Authors · 2022-11-16
> **Thank you for your review**
>
> We thank the reviewer for their time and insightful comments on our work. We believe that some of the concerns raised here are due to misunderstandings and we hope that our responses help clarify them thoroughly. We invite the reviewer to further raise any outstanding concerns that would prevent them from increasing their score and we would be happy to address them immediately.
>
> > The meaning of proposed metric embedding variability (EV) is similar to the TDAP constraint. So the value of EV should be small intuitively under TDAP constraint. It is better to provide the explanation and comparison to other existed metrics.
>
> Thank you for this question. Embedding Variability is not directly related to the TDAP constraint as the TDAP constraint is applied to the adjacency matrices and not the embeddings. One of the implications of the TDAP constraint, as we show in Equation 4, is that the rate of embedding change is preserved after perturbations. However, the constants are model-specific and need to be analyzed empirically to see how reasonably low the difference in the embeddings is. For this reason, we present results on the EV metric. The EV metric is inspired by the anomaly scoring methods of the embedding-based anomaly detectors in the literature [1] (reference number based on the list below). Thus, a low EV implies ineffectiveness of the embedding-based anomaly detectors to detect our perturbations. We have added an extended discussion on the anomaly detection methods for TDAP constraint in Appendix L. Please refer to this section for more details on existing anomaly detectors and how this relates with the Embedding Variability. In addition, we add results on a clustering-based anomaly detector based on [2] as well, in Appendix L.1. We have also added a 2-sample t-test to compare $d Z$ before and after perturbation, that complements our results for Embedding Variability well.
>
> > Table 2 only reports the EV metric for TD-PGD, I would like to see more attack methods in the results. And the proposed method seems cannot achieve the best results in appendix H.4, whiy.
>
> We show the results of other methods in Figures 6 and 7 in Appendix H.3. Here, we present comparative results of all methods for $\epsilon = 0.5$ in tabular form (as in Table 2).
>
> | Dataset | Model | TD-PGD | Greedy | Degree | Random |
> | - | - | - | - | - | - |
> | Radoslaw | DySAT | 0.037 (0.177) | 0.044 (0.295) | 0.152 (0.602) | 0.119 (0.369) |
> | | EvolveGCN | 0.452 (1.292) | 0.065 (0.394) | 0.145 (0.645) | 0.126 (0.480) |
> | | GC-LSTM | 0.179 (1.156) | 0.147 (1.674) | 0.295 (2.409) | 0.219 (1.817) |
> | UCI | DySAT | 0.138 (0.252) | 0.218 (0.303) | 0.663 (0.737) | 0.174 (0.280) |
> | | EvolveGCN | 0.225 (0.353) | 0.129 (0.339) | 0.509 (0.806) | 0.172 (0.387) |
> | | GC-LSTM | 0.354 (0.470) | 0.114 (0.189) | 0.676 (0.637) | 0.168 (0.211) |
> | Reddit | DySAT | 0.137 (0.451) | 0.119 (0.404) | 1.283 (1.520) | 0.192 (0.418) |
> | | EvolveGCN | 0.807 (6.366) | 1.299 (17.686) | 4.224 (37.912) | 1.326 (17.698) |
> | | GC-LSTM | 0.255 (0.593) | 0.098 (0.381) | 1.539 (1.980) | 0.135 (0.389) |
>
> In Appendix H.4, we study attack performance on node classification over random targets. In the main paper, we presented node classification results for top-degree targets. Top-degree targets may be more interesting to misclassify from a security standpoint [3] and our method is shown to perform considerably well for these targets. Since DBLP is an attributed dataset, nodes have features that can inform a model’s label prediction. Thus, structural changes may not be effective when the model is using node features for prediction. To this end, we also do feature perturbations by noting that the TDAP constraint can be easily extended to features as well. TD-PGD can also be directly used to generate continuous feature perturbations without the final rounding step. We find, in Figure 9, that TD-PGD with feature perturbations can bring down the performance by $\sim 30$% across different models over the same random targets.
>
> > Some baselines should be considered, e.g., TGA (Chen et al., 2021b).
>
> As mentioned in Section 3.2 in the description of the Greedy method, our Greedy method is exactly the TGA method adapted for our novel TDAP constraint. The original TGA is based on a global budget constraint but as we adapt their approach for our local TDAP constraint for fair comparison, following Algorithm 2, we named it Greedy in the paper. For more clarity, we have updated the paper to replace ‘Greedy’ with “TGA$(\epsilon)$’’.
>
> We also note that a direct comparison between the original TGA and TD-PGD would be unfair to TGA. This is because the perturbations introduced by TGA may not satisfy the TDAP constraint as the perturbations introduced by TGA are only constrained by a global budget, i.e., $\sum_t \lVert A'_t - A_t \rVert \le B$.
>
> Thus,  for a fair comparison with TD-PGD, we adapt TGA to satisfy the TDAP constraint $ \lVert A’_t - A_t \rVert \le \epsilon d A_t$ for all t.

---

> > ### Author Response · Authors · 2022-11-16
> > **References for the above responses**
> >
> > [1] Xu, Kaidi, et al. "Topology attack and defense for graph neural networks: An optimization perspective." arXiv preprint arXiv:1906.04214 (2019).
> >
> > [2] Geisler, Simon, et al. "Robustness of graph neural networks at scale." Advances in Neural Information Processing Systems 34 (2021): 7637-7649.

---

> ### Author Response · Authors · 2022-11-18
> **Requesting feedback**
>
> Thank you once again for your review. We have tried to carefully address your concerns in our responses and it would be very valuable to us if you could provide your feedback. If any issues still remain that need to be resolved, please let us know.

---

> ### Author Response · Authors · 2022-12-02
> **Requesting feedback**
>
> Thank you once again for your review. We have tried to carefully address your concerns in our responses and it would be very valuable to us if you could provide your feedback as soon as possible. If any issues still remain that need to be resolved, please let us know.

---

### Official Review · Reviewer_z1Xm · 2022-10-27

**Confidence:** 3
**Correctness:** 3
**Technical Novelty And Significance:** 3
**Empirical Novelty And Significance:** 3
**Recommendation:** 6

**Clarity, Quality, Novelty And Reproducibility:**

The technical details are presented clearly. However the experiment section needs to be strengthed

**Strength And Weaknesses:**

Strength:
* The paper focuses on a new perspective of dynamic graph attack
* The draft is organized well and easy to follow
* The technical details are presented clearly

Weakness:
* The experiments results need more analysis and to be better presented; some experiment design needs to be better justified
* The motivation of proposing the new constraint need to be highlighted

**Summary Of The Paper:**

In this paper, the authors propose a new constraint on generating perturbations on discrete dynamic graphs and a white-box attack methods on dynamic graphs. Several experiments have been conducted to verify the effectiveness of the proposed method.

**Summary Of The Review:**

My major concern is on the evaluation section:

- It would be better if the authors can include more state-of-the-methods on dynamic graph modeling in the evaluation (e.g., ROLAND) in addition to the current methods (e.g., DySAT, etc.) and discuss whether the proposed attacking method apply to them or not
- The relative drop figure can be misleading (e.g., Fig. 1) as the y-axis scales are not aligned. I suggest the authors to plot the absolute values instead
- Some design choices need to be better justified: e.g., why only selecting top degree nodes to evaluate the classification tasks
- I suggest the authors to conduct more analysis instead of just presenting the results: e.g., DySAT seems suffering less from the proposed attack methods. What is the potential reason for this robustness compared w/ other mechanisms? Any specific mechanisms can be further designed to attack this category of modeling algorithms?

---

> ### Author Response · Authors · 2022-11-16
> **Thank you for your review**
>
> We thank the reviewer for their positive feedback and constructive suggestions on improving the evaluation section of the paper that we find very valuable towards achieving a high quality presentation of the paper. Below we address all the concerns raised by the reviewer and invite them to raise any outstanding concerns that would prevent them from increasing their score. We would be very happy to engage in an iterative discussion towards improving the presentation of paper while allaying any technical concerns.
>
>
> > The experiments results need more analysis and to be better presented; some experiment design needs to be better justified
>
> We have attempted to improve the presentation of the evaluation section in the revised version of the paper and provided more results, justifications and support for the claims made in the paper. We also discuss these improvements in response to your questions below.
>
> > The motivation of proposing the new constraint need to be highlighted
>
> In the literature, a budget constraint has been widely used to enforce imperceptibility in graphs [1,2] (reference number based on the list below). However, this constraint only bounds the total amount of perturbations that can be introduced by an attacker. When the input is dynamic, as is the case for dynamic graphs, the perturbations should be constrained in the context of how the input evolves. The budget constraint completely ignores the graph dynamics and could lead to a drastic change in the evolution trend of the graph (for example, by making all the perturbations at the final time step) and thus, such attacks can be easily detected.
>
> Thus, we present a constraint that can ensure that the introduced perturbations do not disrupt the evolving trend of the dynamic graphs. We show theoretically that perturbations made under the proposed constraint preserve the trend of structural (Theorems 1 and 4 (Appendix K)) and embedding evolution (Theorems 2, and 5, 6 (Appendix K)). In the revised version, we also discuss and show empirically how TDAP-constrained perturbations are effective against different dynamic graph anomaly detectors (please see Appendix L and [responses to reviewer kVti](https://openreview.net/forum?id=yUY15QBERj&noteId=POlv9qh9Smu)). Such a constraint is thus highly desirable as it shows that dynamic graph representation models can be attacked while theoretically (and empirically) evading detection by anomaly detectors.
>
>
> > It would be better if the authors can include more state-of-the-methods on dynamic graph modeling in the evaluation (e.g., ROLAND) in addition to the current methods (e.g., DySAT, etc.) and discuss whether the proposed attacking method apply to them or not
>
> We would like to note that ROLAND [3] was released publicly on August 15, 2022 at ACM SIGKDD 2022, i.e., only one month prior to the submission deadline for ICLR to the best of our knowledge. As per the ICLR guidelines [here](https://iclr.cc/Conferences/2023/ReviewerGuide), if a paper was published (i.e., at a peer-reviewed venue) on or after May 28, 2022, authors are not required to compare their own work to that paper.  Since ROLAND is a new model and we did not have the time to run all the experiments on it prior to the submission deadline, our extensive study on multiple models (DySAT, EvolveGCN, GC-LSTM) provides a comprehensive overview of the effectiveness of our attack.
>
> Having said that, we thank the reviewer for pointing us to this recent method.  To address the reviewer’s last point, TD-PGD is independent of the victim model’s architecture (refer to Algorithm 1). Also, we verified the ROLAND model and we do not find any technical limitation on the applicability of TD-PGD to ROLAND. Consequently, we have started experiments for  ROLAND as the victim model. While we will make every effort to include them in the revision before the end of the discussion period, it is very likely that we may not have the results by then due to limited time. If that is the case, we will add the results to the final paper version. That said, we request the reviewer to evaluate the paper’s merit per the reviewer guidelines of ICLR 2023 against baselines published prior to the cutoff date.

---

> > ### Author Response · Authors · 2022-11-16
> > **Addressing design and empirical concerns**
> >
> > > The relative drop figure can be misleading (e.g., Fig. 1) as the y-axis scales are not aligned. I suggest the authors to plot the absolute values instead
> >
> > Relative drop allows one to compare the performance of the attack methods across different models, which is harder to do with absolute values. Since the objective of the attacker is to drop the performance, it is easier to evaluate the efficacy of an attack method by noting the drop in performance instead of the performance metric itself. Furthermore, we plot the relative drop instead of just the drop in performance so that it becomes easier to compare if the drop is significant enough across models regardless of whether the original performance was high or not. For example, consider two models with original performance as $100$% and $50$% and the attacker drops the performance by $10$% in each of them. It can be argued then that the drop from $50$% to $40$% is more significant than dropping from $100$% to $90$%. This is captured using a relative drop which would be $20$% for model 2 and $10$% for model 1.
> >
> > We use relative drop instead of absolute values to make it easier to assess the efficacy of the attack methods across different models. While the y-axis scales across figures are not aligned for the relative drop, we believe that it still provides a more useful visualization in assessing the efficacy of the attacker. From the reviewer’s current comments, we do not see how relative drop could be misleading so it would be helpful if the reviewer can elaborate on a specific confusion with regard to this metric.
> >
> > > Some design choices need to be better justified: e.g., why only selecting top degree nodes to evaluate the classification tasks
> >
> > As noted in Section 5.2, we present results for top-degree targets in the main paper and defer the results for random targets to the Appendix H.4. We believe that misclassifying the labels for influential top-degree targets can significantly impact a model's usability in practice. These design choices are in line with other attack papers on graphs [2]. We would be happy to switch the presentation and instead, present random targets in the main paper (and top-degree in the Appendix) if the reviewer feels that choice to be more valuable.
> >
> > > I suggest the authors to conduct more analysis instead of just presenting the results: e.g., DySAT seems suffering less from the proposed attack methods. What is the potential reason for this robustness compared w/ other mechanisms? Any specific mechanisms can be further designed to attack this category of modeling algorithms?
> >
> > A key difference of DySAT compared to other models is the use of temporal and structural attention mechanisms. Robustness of attention mechanism to adversarial perturbations has been studied in other domains [4-6]. It has been found that attention-based models are more robust to perturbations than other recurrent baselines in NLP. Since DySAT involves self-attention on graph structure and temporal dynamics, its observed robustness is in line with existing results of the robustness of self-attention networks in the NLP literature [4]. [4] also provide a theoretical analysis of why a self-attentive model would be more robust than a recurrent one. Due to the sequential nature of the discrete-time dynamic graph data, these theoretical justifications directly transfer to justify the superior robustness of DySAT as compared to GC-LSTM and EvolveGCN. Simultaneously, we would also like to note that attention mechanisms have also been used in the literature to design defense mechanisms against static graph attacks [7-9]. These rely on learning attention mechanisms that can distinguish between adversarial and clean edges so that adversarial edges contribute less to the prediction.
> >
> > To address the reviewer's last question, one way to counter such a defense could be to develop adaptive attacks against attention-based approaches [10]. Self-attention used in DySAT can be seen as an implicit defense and one can follow guidelines from [10] to construct adaptive attacks. The first step noted by them is to come up with an improved loss function on which gradient descent can generate adversarial samples. This is followed by a method to minimize that loss function. Thus, one way to attack methods like DySAT could be to use TD-PGD on an improved loss function that can incorporate the attention weights in a differentiable manner. We intend the development of such methods as future work.

---

> > > ### Author Response · Authors · 2022-11-16
> > > **References for the above responses**
> > >
> > > [1] Dai, Hanjun, et al. "Adversarial attack on graph structured data." International conference on machine learning. PMLR, 2018.
> > >
> > > [2] Chen, Jinyin, et al. "Time-aware gradient attack on dynamic network link prediction." IEEE Transactions on Knowledge and Data Engineering (2021).
> > >
> > > [3] You, Jiaxuan, Tianyu Du, and Jure Leskovec. "ROLAND: graph learning framework for dynamic graphs." Proceedings of the 28th ACM SIGKDD Conference on Knowledge Discovery and Data Mining. 2022.
> > >
> > > [4] Hsieh, Yu-Lun, et al. "On the robustness of self-attentive models." Proceedings of the 57th Annual Meeting of the Association for Computational Linguistics. 2019.
> > >
> > > [5] Mu, Norman, and David Wagner. "Defending against adversarial patches with robust self-attention." ICML 2021 Workshop on Uncertainty and Robustness in Deep Learning. 2021.
> > >
> > > [6] Vaishnavi, Pratik, et al. "Can attention masks improve adversarial robustness?." International Workshop on Engineering Dependable and Secure Machine Learning Systems. Springer, Cham, 2020.
> > >
> > > [7] Zhu, Dingyuan, et al. "Robust graph convolutional networks against adversarial attacks." Proceedings of the 25th ACM SIGKDD international conference on knowledge discovery & data mining. 2019.
> > >
> > > [8] Zhang, Xiang, and Marinka Zitnik. "Gnnguard: Defending graph neural networks against adversarial attacks." Advances in neural information processing systems 33 (2020): 9263-9275.
> > >
> > > [9] Tang, Xianfeng, et al. "Transferring robustness for graph neural network against poisoning attacks." Proceedings of the 13th international conference on web search and data mining. 2020.
> > >
> > > [10] Tramer, Florian, et al. "On adaptive attacks to adversarial example defenses." Advances in Neural Information Processing Systems 33 (2020): 1633-1645.

---

> ### Author Response · Authors · 2022-11-18
> **Requesting feedback**
>
> Thank you once again for your review. We have tried to carefully address your concerns in our responses and it would be very valuable to us if you could provide your feedback. If any issues still remain that need to be resolved, please let us know.

---

> ### Author Response · Authors · 2022-12-02
> **Requesting feedback**
>
> Thank you once again for your review. We have tried to carefully address your concerns in our responses and it would be very valuable to us if you could provide your feedback as soon as possible. If any issues still remain that need to be resolved, please let us know.

---

### Official Review · Reviewer_gWpZ · 2022-10-31

**Confidence:** 3
**Correctness:** 3
**Technical Novelty And Significance:** 4
**Empirical Novelty And Significance:** 3
**Recommendation:** 6

**Clarity, Quality, Novelty And Reproducibility:**

- Clarity: Medium. The paper is mostly well written, but some important points are unclear (see my questions above), and there are some notation issues.
- Quality: Medium. There are potentially some technical issues with some experiment results, but the proposed TDAP constraint and TD-PGD methods seem mostly sound.
- Novelty: High. There is both high methodological and empirical novelty in this paper. The TDAP constraint and TD-PGD attack method constitutes the methodological novelty, while the results on attack vulnerability for the three different GNNs constitutes the empirical novelty that may be useful beyond this paper.
- Reproducibility: Medium. Lots of details on experiment setup are provided in the appendix, along with code. I skimmed through some of the code but did not run it. Some of the results seem implausible, so I don't know how reproducible they would be.


**Strength And Weaknesses:**

Strengths:
- Proposed TDAP constraint and projected gradient descent relaxation to find near-optimal attacks are principled and technically sound.
- Proposed Temporal Dynamics-aware Projected Gradient Descent (TD-PGD) attack method is much more effective than greedy and other baseline attacks subject to the same TDAP constraint.
- There does not appear to be much prior research on attacks for dynamic GNNs, so novelty is high.

Weaknesses:
- Some empirical results regarding the proposed Embedding Variability (EV) seem highly implausible, if not impossible.
- Attack performance for online version of TD-PGD is inexplicably poor in several experiments. More investigation should be conducted as to why this is happening, beyond just a conjecture.

Questions:
1. What is the meaning of the notation $\mathbf{S}_t \leq \varepsilon_t$ in line 4 of Algorithms 1 and 3? On one side of the inequality, you have a matrix, and on the other side, you have a scalar. Is there supposed to be $\ell_1$ norm around $\mathbf{S}_t$?
2. Also for line 4 in Algorithms 1 and 3, why randomized rounding using Bernoulli($\mathbf{s}_t^{(N)}$) rather than just directly rounding? Can you guarantee that the discretized $\mathbf{S}_t$ satisfies the TDAP constraint, and not just the continuous $\mathbf{s}_t$?
3. How is a standard deviation of 6.37 for the EV of EvolveGCN possible when the mean is 0.81, and EV must be greater than 0 according to Equation (7)? Similarly, I see error bars going into the negatives in Figure 6.

Typos and minor issues:
- Some minor inconsistencies in notation in Algorithm 1: I believe the vectors $\mathbf{a^{(i)}}$ and $\mathbf{s^{(i)}}$ should also have subscript $t$, and $s_t^{(N)}$ should actually be a vector $\mathbf{s}_t^{(N)}$.
- Abstract and Section 5.1: "upto" -> "up to"


**Summary Of The Paper:**

The authors propose a novel adversarial attack strategy for discrete-time graph neural networks (GNNs). The objective is to perturb the sequence of graphs to decrease accuracy on prediction tasks such as dynamic link prediction and node classification. To preserve temporal dynamics of the graph to avoid detection, the authors propose the temporal dynamics-aware perturbation (TDAP) constraint. They then propose a projected gradient descent algorithm to identify effective perturbations that satisfy the TDAP constraint. They demonstrate strong empirical performance when attacking three different discrete-time GNNs.


**Summary Of The Review:**

The authors present a novel and technically sound approach to attack dynamic GNNs. I have some technical concerns regarding empirical results that prevent me from more strongly recommending acceptance. I should note also that my expertise is in building models for dynamic networks and not attacking them, so my knowledge of research on adversarial learning is lower.

---

> ### Author Response · Authors · 2022-11-16
> **Thank you for your review**
>
> We thank the reviewer for their candid and insightful comments that have helped to improve the quality of our paper. We are glad that the reviewer finds our work highly novel and we appreciate the technical concerns raised for the experiments. We believe that these concerns have arisen partly due to weak presentation on our end (which we have now fixed) and partly due to some misunderstanding (which we have clarified in our explanation). Below, we address these concerns rigorously with specific responses and invite the reviewer to assess the work in light of these responses. We encourage the reviewer to state any further outstanding concerns that would prevent them from increasing the score and we would be happy to address them in the next iteration.
>
> > What is the meaning of the notation St≤εt in line 4 of Algorithms 1 and 3? On one side of the inequality, you have a matrix, and on the other side, you have a scalar. Is there supposed to be ℓ1 norm around St?
>
> Thank you for flagging this. That is a typo. There is supposed to be an $\ell 1$ norm around $\mathbf{S}_t$. Thanks for pointing this out. We have updated the Algorithms in the revised version.
>
> >Some minor inconsistencies in notation in Algorithm 1: I believe the vectors a(i) and s(i) should also have subscript t, and st(N) should actually be a vector st(N). Abstract and Section 5.1: "upto" -> "up to"
>
> Thanks for pointing out the typos. We have fixed them in the updated version. However, we would like to note that $\mathbf{a}^{(i)}$ and $\mathbf{s}^{(i)}$ in the gradient step should not have a $t$ subscript. This is because the loss is a function of $\mathbf{s}^{(i)}$ at all time steps. Thus, we pass the full vector (concatenating all the time steps) to the loss functions and take the gradient step to update the full vector $\mathbf{s}^{(i)}$ and denote the updated vector as $\mathbf{a}^{(i)}$.
>
>
>
> > Also for line 4 in Algorithms 1 and 3, why randomized rounding using Bernoulli(st(N)) rather than just directly rounding? Can you guarantee that the discretized St satisfies the TDAP constraint, and not just the continuous st ?
>
> We used randomized rounding inspired by other works using PGD for graphs [1,2] (reference number based on the list below). This is done instead of just directly rounding so as to ensure that the rounded discrete solution has a chance to satisfy the TDAP constraint. In fact, with a simple example, one can show that a discrete solution obtained from directly rounding does not guarantee that the constraint will be satisfied. To give an example, let us consider a static 1D case with the constraint $s \le \epsilon \cdot x$ with $\epsilon = 0.5$ and $x = 1.5$. Suppose after PGD iterations, we obtain $s = 0.6$. Note that $s = 0.6 \le \epsilon \cdot x = 0.5 \cdot 1.5 = 0.75$. However, if we directly round $s$, we get $1.0 \nleq 0.75$. Thus, the constraint is not satisfied.
> This inspires our use of randomized rounding where the continuous solution is rounded by sampling from a Bernoulli distribution with the PGD solution as probabilities for a fixed number of iterations. We pick the solution that decreases the loss by the most while satisfying the constraint. To ensure faster convergence, we use the top-k heuristic sampling in the first iteration inspired by [2]. Note that a top-k heuristic to discretize $\mathbf{S}_t$ is guaranteed to satisfy TDAP. This is because, at each time step, we select the top $\lfloor \epsilon d \mathbf{A}_t \rfloor$ perturbations based on the probabilities $\mathbf{S}_t$.
>
> Thanks for bringing this up as it has provided us with an opportunity to discuss the relevant missing details from the paper. We have now added a separate section as Appendix M to discuss the randomized rounding algorithm (Algorithm 4) in more detail.

---

> > ### Author Response · Authors · 2022-11-16
> > **Addressing empirical concerns**
> >
> > > How is a standard deviation of 6.37 for the EV of EvolveGCN possible when the mean is 0.81, and EV must be greater than 0 according to Equation (7)? Similarly, I see error bars going into the negatives in Figure 6.
> >
> > Embedding variability measures the relative variability of the consecutive change of the embedding of the target node. We find it by comparing the range of the distance between consecutive embeddings before and after perturbation. Thus, this value can be unbounded in the range $[0, \infty]$. A careful analysis of the distribution of these values for the case mentioned by the reviewer leads us to note that the high standard deviation is caused by a single outlier in the data which has an embedding variability of over 200 while the rest are all below 0.2. For better readability, we had already reported median values (with quantile ranges of 10% and 90%), which we believe is a better metric here (please see Figure 7). The corresponding results for the median (with quantile values at 10% and 90% in the parentheses respectively) of values in Table 1 are given as -
> >
> > | Dataset | Model | EV (median (IQR 10%, IQR 90%)) |
> > | -- | -- | -- |
> > | DBLP | DySAT | 0.096 (0.012, 0.521) |
> > | | EvolveGCN | 0.274 (0.039, 1.207) |
> > | | GC-LSTM | 0.412 (0.068, 1.890) |
> > | Radoslaw | DySAT | 0.002 (0.000, 0.042) |
> > | | EvolveGCN | 0.245 (0.031, 0.786) |
> > | | GC-LSTM | 0.074 (0.012, 0.195) |
> > | UCI | DySAT | 0.057 (0.005, 0.397) |
> > | | EvolveGCN | 0.109 (0.010, 0.619) |
> > | | GC-LSTM | 0.201 (0.027, 0.850) |
> > | Reddit | DySAT | 0.021 (0.000, 0.341) |
> > | | EvolveGCN | 0.131 (0.020, 1.230) |
> > | | GC-LSTM | 0.130 (0.017, 0.843) |
> >
> > We have added this table in Appendix H.3 as well. We would also like to note that the error bars in Figure 6 are drawn symmetrically in both directions since it’s a standard deviation. That is why the error bars can go negative. However, as pointed out by the reviewer, the EV metric as defined in Equation 7 cannot go below 0 and thus, this may not be the most intuitive way to visualize this. Therefore, we had an additional plot as Figure 7 (also in the first version) that plots median values with asymmetrical error bars as 10% and 90\% inter-quantile range values. One can note here that the values are all above $0$ as one may expect.
> >
> > >  Attack performance for the online version of TD-PGD is inexplicably poor in several experiments. More investigation should be conducted as to why this is happening, beyond just a conjecture.
> >
> > We believe that there is some misunderstanding in the interpretation of the results for online settings. Figure 3 shows that online TD-PGD outperforms or achieves comparable performance to other baselines in 7 out of 9 cases while achieving up to $5$x improvement. Further, to the best of our knowledge, this is the first work that supports attacks on dynamic graphs in an online setting, a practical but very challenging setting (as the attacker has to select perturbations at each intermediate timestep without having access to any future snapshots, including the downstream objective at a future time step). This opens new avenues for studying online adversarial attacks for dynamic graphs and we believe that the results of online TD-PGD are promising as a first step.
> >
> > At the same time, to the reviewer's point, our current explanation of the negative results for the two cases (in Figures 3(a) and 3(i)) is based on the gradient ascent dynamics. Online TD-PGD selects the perturbations while following gradient ascent on the classification loss calculated with the final timestep’s edge labels and current timestep’s edge probabilities for the target edge. This may be misaligned as the perturbations found to minimize using the current timestep’s probabilities may be different than when minimizing using the final timestep’s probabilities.
> > We agree that there may be other factors contributing to the performance drop and intend to perform a rigorous study and design of new techniques specific to online settings for future work.
> >
> > > Some empirical results regarding the proposed Embedding Variability (EV) seem highly implausible, if not impossible.
> >
> > Please see our response to Q3. Also, let us know if that question doesn’t cover the stated implausibility and if there were any specific cases in the EV results that you found implausible and we will be happy to address them.
> >
> > **References**
> >
> > [1] Xu, Kaidi, et al. "Topology attack and defense for graph neural networks: An optimization perspective." arXiv preprint arXiv:1906.04214 (2019).
> >
> > [2] Geisler, Simon, et al. "Robustness of graph neural networks at scale." Advances in Neural Information Processing Systems 34 (2021): 7637-7649.

---

> ### Author Response · Authors · 2022-11-18
> **Requesting feedback**
>
> Thank you once again for your review. We have tried to carefully address your concerns in our responses and it would be very valuable to us if you could provide your feedback. If any issues still remain that need to be resolved, please let us know.

---

> > ### Comment · Reviewer_gWpZ · 2022-12-02
> > **Thank you for the responses**
> >
> > Thank you for the responses. Most of my concerns have been clarified or addressed.
> >
> > I disagree with the other reviewers who find the novelty to be limited. I find the presentation quality to be more problematic than the novelty. If this paper is not accepted, I hope to see a revised version with improved presentation accepted at a similar venue in the future.

---

> > > ### Author Response · Authors · 2022-12-02
> > > **Thank you and a question about presentation quality**
> > >
> > > Dear Reviewer gWpZ,
> > >
> > > Thank you for acknowledging our response. We are delighted to see your appreciation about the novelty of this work. We wish to mention that, in addition to addressing your concerns in this response, we had uploaded the revised version of the paper with an updated presentation and inclusion of several more details and results. Given this, it would be very helpful to know if there are still any significant outstanding concerns in your mind related to the presentation quality of the revised version that stopped you from increasing your initial score at this point.

---

### Official Review · Reviewer_kVti · 2022-11-03

**Confidence:** 3
**Correctness:** 3
**Technical Novelty And Significance:** 2
**Empirical Novelty And Significance:** 2
**Recommendation:** 5

**Clarity, Quality, Novelty And Reproducibility:**

The method and the research problem are not surprising. Writting is good. Novelty is limited.

**Strength And Weaknesses:**

Strength:
1. The paper is well written and easy to follow.
2. Theoretical proofs are given.

The author proposed TD-PGD as an attack approach on dynamic graph models. I have the following questions that are mainly related to the proposed constraints.

1. Limiting the impact of attack perturbations on the change rate of structures and embeddings sounds reasonable, but it would be conditional. Nowadays, spatial- and spectral-based anomaly detection techniques on graphs are developing rapidly. The attack constraint on dynamic graphs is fundamental to this task, thus the authors should give a rigorous proof about why and when anomaly detection algorithms fail in the face of the proposed constraints.
2. The proof is not rigorous. First, although the bound is proven, there is no limit to the constants $\alpha$ and $\beta$, which might seriously affect the average change preservation. Second, the change of a matrix is directional, it is unreasonable to just consider the norm of average changes. To my knowledge, spectral anomaly detection methods can effectively capture the anomaly caused by the change directions.
3. The proof is inconsistent with the target. It is not clear how the proven bound echoes the word ‘preserves’. A small enough $\overline{dA’}$ instead limits the dynamics of dynamic graphs, which increases risk being detected. I think at least the following formula should be proved:  $\vert \overline{dA’}-\overline{dA} \vert \leq \phi$, where $\phi$ is the bound to make the perturbed structure change rate preserved. ‘Implications’ are intuitively sound, but theoretical proof might be flawed.
4. The authors are suggested to empirically demonstrate that anomaly detection is insensitive to the proposed attack method. If the attack itself is easily detected, the performance drop may be ineffective.


**Summary Of The Paper:**

This paper proposes a novel perturbation constraint (named TDAP) for adversarial attack on discrete-time dynamic graph. Importantly, TDAP can preserve the average rate of structure and embedding changes. As TDAP constraint has not been studied before for dynamic graph, authors propose a theoretically grounded approach named TD-PGD for easily extending TDAP to find attacks in a more practical online setting.
Experiments are conducted on dynamic link prediction and node classification tasks to show the effectiveness of TD-PGD.


**Summary Of The Review:**

The paper is well written in terms of grammar and writing style. Theoretical proofs are not mathematically rigorous and are inconsistent with the constraint targets. Necessary assumptions are not made clear.

---

> ### Author Response · Authors · 2022-11-16
> **Thank you for your review**
>
> We thank the reviewer for their time and efforts in providing a detailed assessment of our paper. We are delighted to see the enthusiastic questions regarding the effectiveness of our approach in the presence of anomaly detection algorithms. We note that the main focus of our work is to build an approach that can perturb graphs such that it preserves the embeddings produced by representation learning methods (victim models) within a non-detectable range while decreasing the performance of downstream tasks that use these representations. This is in line with recent directions of performing adversarial attacks on static graphs [17, 18] (reference number based on the list below) that hamper the performance of representation learner (a key component found in most real-world applications expanding systems such as recommendation systems [19] and QA systems [20]). As we discuss below, our approach will naturally be most effective against neural approaches to anomaly detection that uses changes in graph representations (induced by graph perturbations) to detect anomalies.  Having said that, the questions from the reviewer have encouraged us to perform an in-depth study on the broader class of existing approaches for anomaly detection and discuss (plus assess theoretically and empirically where feasible) the effectiveness of our approach in presence of such detection mechanisms. We discuss this in detail below and also in our revised version.
>
> Before presenting detailed responses, we summarise the revisions we have added to the paper here-
> 1. We have added an extended discussion in Appendix L to study the effectiveness of TDAP-constrained perturbations in presence of anomaly detection algorithms.
> 2. We have added Figure 14 in Appendix L.1 that plots Netwalk’s anomalous scores [10] before and after perturbation for different datasets and victim graph representation models.
> 3. Table 6 (in Appendix L.1) is added to show p-values of a 2-sample t-test between $d \mathbf{Z}$ and $d \mathbf{Z}’$ for different model-dataset pairs at different epsilons.
> 4. We prove Theorem 4 in Appendix K to give a lower bound on $\overline{d \mathbf{A}’}$ in terms of $\overline{d \mathbf{A}}$.
> 5. We prove Theorem 6 in Appendix K to give a lower bound on $\overline{d \mathbf{Z}’}$ in terms of $\overline{d \mathbf{Z}}$ as well.
>
> In our specific responses below, we first address the comments related to the anomaly detection algorithms. Next, we clarify a few points regarding our theoretical proofs and also present revised/extended versions of the proofs where applicable. To address the novelty concerns raised by the reviewer, we conclude our responses with a re-emphasis on the key contributions of this work. We have also included this material in the revised version of our paper and cited the appropriate sections in the discussion below.  We invite the reviewer to evaluate our changes and discuss any outstanding concerns that would prevent them from increasing the score. We will be happy to address them rigorously in the next iteration.

---

> > ### Author Response · Authors · 2022-11-16
> > **TD-PGD and Anomaly Detection Algorithms (1/2)**
> >
> >
> > > Limiting the impact of attack perturbations on the change rate of structures and embeddings sounds reasonable, but it would be conditional. Nowadays, spatial- and spectral-based anomaly detection techniques on graphs are developing rapidly. The attack constraint on dynamic graphs is fundamental to this task, thus the authors should give a rigorous proof about why and when anomaly detection algorithms fail in the face of the proposed constraints.
> >
> > As rightly pointed out by the reviewer, there has been a rapid increase in the development of anomaly detection techniques for graphs. At the same time, existing approaches [1-3], including the state-of-the-art [4], that studies spectral properties of the graphs for anomaly detection are largely limited to static graphs and not applicable in the context of this work on dynamic graphs. Hence, we focus on anomaly detection techniques for dynamic graphs to address the raised concerns.
> > Specifically, inspired by the surveys on this topic [5-7], there are two broad categories of anomaly detection methods for dynamic graphs: supervised and unsupervised methods.
> >
> > 1. **Unsupervised methods** detect anomalies by studying certain properties of the dynamic graph. These methods are applicable to the setting in our paper and hence we focus our detailed discussion and analysis on unsupervised approaches.
> > Such approaches can be further divided into two types:
> >    - Traditional approaches that study different graph distance metrics between consecutive snapshots, for e.g., graph edit distance, hamming distance, etc. [8,9]. If the distance with the previous snapshot exceeds a threshold, then the instance is deemed anomalous. These works, however, focus on effectively detecting graph anomalies instead of edge anomalies at any point in time. On the other hand, we consider a targeted attack setting, where perturbations (anomalies) are added to only change the local behaviour around the target and not the global graph dynamics. A graph-level anomaly detection would thus not be able to detect local perturbations. For this reason, we study these distance metrics for just the ego-network around the target node. Theorems 1 and 4 (in Appendix K) shows that the average rate of structural change remains preserved, which implies effectiveness against such threshold based anomaly detectors (one of the most prevalent kinds of detectors for dynamic graphs [6]. Suppose one had chosen a threshold $B$ such that $d \mathbf{A}_t \geq B$ is defined as an anomaly. Initially, if the graph sequence was not anomalous on average, i.e., $\overline{d \mathbf{A}} \le B$, then after perturbation, $\overline{d \mathbf{A}’} \geq B$ would happen if $\overline{d \mathbf{A}’} \geq |1 - 2 \epsilon| \overline{d \mathbf{A}} \geq B$, i.e., $ \overline{d \mathbf{A}} \geq B/(| 1 - 2\epsilon |)$. Thus, one can choose a value of $\epsilon$ such that this does not hold for a given threshold $B$ and snapshot dynamics $\overline{d \mathbf{A}}$, thereby evading detection.
> >
> >     - Neural approaches that employ embedding-based strategies for anomaly detection. We consider the state-of-the-art NetWalk [10] and DynGEM [11] approaches under this type.
> >
> >       **NetWalk**[10] clusters the embeddings of a sampled edge set at each timestep and flags an edge to be anomalous if its distance in the embedding space from each cluster exceeds a threshold. TDAP-constrained solutions would be effective against such a detector since as noted in Theorem 5 (Appendix K), embeddings before and after perturbation change only by a small factor of the actual change in the adjacency matrices relative to the previous timestep. Thus, the distance of edge embeddings from the cluster centres would change according to the structural evolution at that time.
> >       **DynGem** [11], uses the norm distance between embeddings in consecutive snapshots to find anomalies when the distance exceeds a threshold. Theorems 2 and 6 directly note that the embedding distance between consecutive snapshots would be preserved. We also show empirical evidence of this preservation via our metric Embedding Variability. Note that Embedding Variability measures the ratio of the range of the consecutive embedding distance before and after the perturbation. Thus, if the embedding variability is $\kappa$, it means the range of $d Z_{\tau}$ after perturbation is $1 + \kappa$ (or $1 - \kappa$) times the range of $d Z_{\tau}$ before perturbation. If the threshold was set to be $D$, then our perturbation would be anomalous if $D \geq d Z_{\tau} \geq D/ (1 + \kappa)$. Thus, we can choose an $\epsilon$ such that the EV that we get (i.e., $\kappa$) does not satisfy the above inequality for a given threshold $D$ and embedding evolution $d Z_{\tau}$. In this work, we show how $\kappa$ varies with different $\epsilon$ and can be as low as $0.04$ even for an $\epsilon = 0.5$.

---

> > > ### Author Response · Authors · 2022-11-16
> > > **TD-PGD and Anomaly Detection Algorithms (2/2)**
> > >
> > >
> > > 2. **Supervised methods** train a model with supervised labels for anomalous edges. For example, [12,13] use Cross Entropy loss; [14,15] use margin loss on a parameterized score calculated for each edge using node embeddings. A key requirement of these approaches is the availability of supervisory signals for training the detectors. As we do not assume the availability of such supervision (and hence not accessible to the attacker), these types of detection algorithms are not applicable to our setting.
> > >
> > >
> > > We add this as an extended discussion with results in Section L of the Appendix.
> > >
> > >
> > > > The authors are suggested to empirically demonstrate that anomaly detection is insensitive to the proposed attack method. If the attack itself is easily detected, the performance drop may be ineffective.
> > >
> > > While the theoretical justification discussed before are sufficient to demonstrate the efficacy of our approach against traditional approaches,  this may not be the case for neural approaches. To that end, below we discuss and present the effectiveness of our approach to evade detection in presence of both DynGEM and NetWalk, state-of-art neural approaches for anomaly detection on dynamic graphs.
> > >
> > > **Results on DynGEM.** As noted in the revised version (Appendix L and earlier, in Section 4 of the main paper), Embedding Variability is a reasonable metric to test against embedding-based anomaly detectors that rely on distance between embeddings of consecutive snapshots. DynGem  relies on a threshold that can be specific to datasets and tuned according to real anomalies. Thus, we measure the closeness between the distributions of the consecutive difference in the embeddings before and after perturbation. In particular, we consider how different the range of the consecutive distance is before and after perturbation and show its values in Table 2 (in Section 5), Figures 6 and 7 (in Appendix H). One can note that in the majority of cases, the attack methods have a value close to zero. In addition, we also conducted a 2-sample t-test between the consecutive embedding distance before and after perturbation and found that the distance is significantly similar in $70$ % of the cases. Please refer to Appendix L.1 and Table 6 for more details.
> > >
> > > **Results on NetWalk.** We also empirically demonstrate the insensitivity of NetWalk to our attacks. In particular, we calculate the anomalous score of NetWalk before and after perturbation as calculated through trained embeddings of different victim models. Here, we present the difference between the anomalous score (i.e, the distance from the nearest cluster center) before and after perturbation by TD-PGD.
> > >
> > > | Model | Dataset | Epsilon | Anomalous Diff (Median (IQR10, IQR90)) |
> > > | ------ | ---- | ---- | ---- |
> > > | GCLSTM | Radoslaw | 0.02 | 0.01 (0.0, 0.05) |
> > > | GCLSTM | Radoslaw | 0.5 | 0.04 (0.01, 0.13) |
> > > | GCLSTM | Radoslaw | 0.9 | 0.04 (0.01, 0.12) |
> > > | GCLSTM | UCI | 0.02 | 0.0 (0.0, 0.03) |
> > > | GCLSTM | UCI | 0.5 | 0.02 (0.0, 0.13) |
> > > | GCLSTM | UCI | 0.9 | 0.02 (0.0, 0.2) |
> > > | EvolveGCN | Radoslaw | 0.02 | 0.0 (0.0, 0.05) |
> > > | EvolveGCN | Radoslaw | 0.5 | 0.14 (0.01, 0.59) |
> > > | EvolveGCN | Radoslaw | 0.9 | 0.16 (0.01, 0.74) |
> > > | EvolveGCN | UCI | 0.02 | 0.0 (0.0, 0.0) |
> > > | EvolveGCN | UCI | 0.5 | 0.0 (0.0, 0.01) |
> > > | EvolveGCN | UCI | 0.9 | 0.0 (0.0, 0.01) |
> > > | DySAT | Radoslaw | 0.02 | 0.0 (0.0, 0.06) |
> > > | DySAT | Radoslaw | 0.5 | 0.02 (0.0, 0.08) |
> > > | DySAT | Radoslaw | 0.9 | 0.02 (0.0, 0.11) |
> > > | DySAT | UCI | 0.02 | 0.0 (0.0, 0.01) |
> > > | DySAT | UCI | 0.5 | 0.01 (0.0, 0.03) |
> > > | DySAT | UCI | 0.9 | 0.01 (0.0, 0.04) |
> > >
> > > One can observe that the difference in the distance from the centroid before and after perturbation is close to zero across all cases. Furthermore, we also conducted a 2-sample t-test between original and perturbed anomalous scores at each epsilon value for each model-dataset pair. The hypothesis of the two distributions being similar was accepted (i.e., $p$-value $> 0.05$) in all but five cases. These were EvolveGCN for Radoslaw at $\epsilon = 0.3, 0.5, 0.7, 0.9$ and DySAT for Radoslaw at $\epsilon = 0.9$. Thus, Netwalk-based anomaly detection would be largely ineffective in capturing TDAP-constrained perturbations. For more details, please refer to Appendix L.1.

---

> > > > ### Author Response · Authors · 2022-11-16
> > > > **Addressing Theoretical concerns**
> > > >
> > > > > The proof is not rigorous. First, although the bound is proven, there is no limit to the constants α and β, which might seriously affect the average change preservation. Second, the change of a matrix is directional, it is unreasonable to just consider the norm of average changes. To my knowledge, spectral anomaly detection methods can effectively capture the anomaly caused by the change directions.
> > > >
> > > > We refer the reviewers to Appendix C, where we discuss the exact forms of the constants $\alpha$ and $\beta$. In particular, $\alpha = 2 \epsilon$ and $\beta = \frac{1}{T} \sum_t {(\lVert A_t\rVert + \lVert A_{t-1}\rVert)}$.
> > > > We also discuss how such a bound is useful compared to a simple budget constraint (only other existing solution for attacks on dynamic graphs before our work). A simple budget constraint gives no bound on the average structural change in terms of the original change. This is because, $\sum_t \lVert S_t \rVert \le \mathcal{B}$, where $\mathcal{B}$ is the budget. Thus, using similar calculations as Theorem 1, we get $d \mathbf{A}’_t \le \frac{2}{T} \mathcal{B} + \frac{2}{T} \sum_t \lVert\mathbf{A}_t\rVert$. Therefore, perturbations introduced using a budget constraint can exceed the average trend of structural change when $\mathcal{B}/T > \epsilon d \mathbf{A}_t$.
> > > >
> > > > As pointed out by the reviewer, TDAP attacks only bind the norm of the changes and do not preserve change in a directional sense. We note here that none of the existing anomaly detection approaches on graphs have looked at the direction of the dynamic graph evolution [5,6]. As noted before and in extended discussion in Appendix L, anomaly detection methods focus on distance between consecutive snapshots, which should be distinguished from the directional distance or “displacement”. Spectral anomaly detectors for graphs are based on spectral distance between consecutive snapshots, defined as $\sum_i | \lambda_i (A_t)- \lambda_i (A_{t-1}) |$ [9] or a Kruglov distance on the eigenvectors [16]. One can note that neither of these gives any directional sense as both $| \lambda_i (A_t) - \lambda_i (A_{t-1}) |$ and Kruglov distances are, by definition, symmetric and thus do not enforce an order between $A_t$ and $A_{t-1}$. To the best of our knowledge, existing spectral anomaly detection methods for dynamic graphs do not capture directional change and we believe that our work can inspire new avenues towards the development of such anomaly detection approaches. Having said that, we would be happy to analyze any anomaly detection method in more detail if the reviewer can cite the approaches that they think are capable of capturing directions, and thereby potentially applicable to our setting.
> > > >
> > > > > The proof is inconsistent with the target. It is not clear how the proven bound echoes the word ‘preserves’. A small enough dA′― instead limits the dynamics of dynamic graphs, which increases risk being detected. I think at least the following formula should be proved: |dA′―−dA―|≤ϕ, where ϕ is the bound to make the perturbed structure change rate preserved. ‘Implications’ are intuitively sound, but theoretical proof might be flawed.
> > > >
> > > > We agree with the reviewer that an upper bound may not be enough to satisfy the claims of preservation. Thus, upon reflection based on the reviewer’s comments, we now prove an additional theorem (Theorem 4) about the lower bound on $\overline{d \mathbf{A}’}$ that can be found in Appendix K. In particular, we prove that $\overline{d \mathbf{A}’} \geq | 1 - 2\epsilon | \overline{d \mathbf{A}}$. Thus, combining with Theorem 1, one can say that $| 1 - 2\epsilon | \overline{d \mathbf{A}} \le \overline{d \mathbf{A}’} \le 2 \epsilon  \overline{d \mathbf{A}} + \beta$. This is closer to the desired equation by the reviewer but with an additional scaling factor on $\overline{d \mathbf{A}}$.
> > > >
> > > > We further use Theorem 4 to prove a lower bound on $d Z$ as well (please refer to Theorem 6 in Appendix K). We thus get $ \chi d Z \le d Z’ \le \gamma d Z + \delta $.

---

> > > > > ### Author Response · Authors · 2022-11-16
> > > > > **Addressing Novelty Concerns**
> > > > >
> > > > > As highlighted in Table 1 of our paper, this is the first work to study **targeted**, **white-box**, **evasion** attacks on **dynamic** graph models. In this paper, we make significant advancements in this novel setting on three fronts. (1) Ours is the first work that studies temporal dynamics aware adversarial attacks for dynamic graphs, i.e., attacks that can theoretically preserve graph evolution. (2) We are the first to study online adversarial attacks for dynamic graph models, a highly practical but challenging setting since the attacker perturbs the graphs in real-time without having access to future snapshots and without changing the past snapshots. (3) Based on new experiments conducted in response  to the reviewer’s comments, we also empirically show the effectiveness of our temporal dynamics-aware attacks against being undetected by anomaly detection algorithms. Such an analysis has never been conducted for adversarial attacks on dynamic graphs and we believe that our contributions will help the community study the robustness of state-of-the-art dynamic graph representation models in practical settings.
> > > > > We, therefore, believe that this is the first principled and state-of-art approach for an important and challenging research topic of attack on dynamic graphs, that is still in its infancy and has great potential avenues for further research.
> > > > >
> > > > >
> > > > >
> > > > >
> > > > > **References**
> > > > >
> > > > > [1] Liu, Yang, et al. "Pick and choose: a GNN-based imbalanced learning approach for fraud detection." Proceedings of the Web Conference 2021. 2021.
> > > > >
> > > > > [2] Dou, Yingtong, et al. "Enhancing graph neural network-based fraud detectors against camouflaged fraudsters." Proceedings of the 29th ACM International Conference on Information & Knowledge Management. 2020.
> > > > >
> > > > > [3] Liu, Zhiwei, et al. "Alleviating the inconsistency problem of applying graph neural network to fraud detection." Proceedings of the 43rd international ACM SIGIR conference on research and development in information retrieval. 2020.
> > > > >
> > > > > [4] Tang, Jianheng, et al. "Rethinking Graph Neural Networks for Anomaly Detection." arXiv preprint arXiv:2205.15508 (2022).
> > > > >
> > > > > [5] Ma, Xiaoxiao, et al. "A comprehensive survey on graph anomaly detection with deep learning." IEEE Transactions on Knowledge and Data Engineering (2021).
> > > > >
> > > > > [6] Akoglu, Leman, Hanghang Tong, and Danai Koutra. "Graph based anomaly detection and description: a survey." Data mining and knowledge discovery 29.3 (2015): 626-688.
> > > > >
> > > > > [7] Ranshous, Stephen, et al. "Anomaly detection in dynamic networks: a survey." Wiley Interdisciplinary Reviews: Computational Statistics 7.3 (2015): 223-247.
> > > > >
> > > > > [8] ​​Bunke, Horst, et al. A graph-theoretic approach to enterprise network dynamics. Vol. 24. Springer Science & Business Media, 2007.
> > > > >
> > > > > [9] Shoubridge, Peter, et al. "Detection of abnormal change in a time series of graphs." Journal of Interconnection Networks 3.01n02 (2002): 85-101.
> > > > >
> > > > > [10] Yu, Wenchao, et al. "Netwalk: A flexible deep embedding approach for anomaly detection in dynamic networks." Proceedings of the 24th ACM SIGKDD international conference on knowledge discovery & data mining. 2018.
> > > > >
> > > > > [11] Goyal, Palash, et al. "Dyngem: Deep embedding method for dynamic graphs." arXiv preprint arXiv:1805.11273 (2018).
> > > > >
> > > > > [12] Cai, Lei, et al. "Structural temporal graph neural networks for anomaly detection in dynamic graphs." Proceedings of the 30th ACM international conference on Information & Knowledge Management. 2021.
> > > > >
> > > > > [13] Zhu, Dali, Yuchen Ma, and Yinlong Liu. "A flexible attentive temporal graph networks for anomaly detection in dynamic networks." 2020 IEEE 19th International Conference on Trust, Security and Privacy in Computing and Communications (TrustCom). IEEE, 2020.
> > > > >
> > > > > [14] Zheng, Li, et al. "AddGraph: Anomaly Detection in Dynamic Graph Using Attention-based Temporal GCN." IJCAI. 2019.
> > > > >
> > > > > [15] Wang, Bin, Teruaki Hayashi, and Yukio Ohsawa. "Hierarchical graph convolutional network for data evaluation of dynamic graphs." 2020 IEEE International Conference on Big Data (Big Data). IEEE, 2020.
> > > > >
> > > > > [16] Shimada, Yutaka, et al. "Graph distance for complex networks." Scientific reports 6.1 (2016): 1-6.
> > > > >
> > > > > [17] Dai, Hanjun, et al. "Adversarial attack on graph structured data." International conference on machine learning. PMLR, 2018.
> > > > >
> > > > > [18] Jin, Wei, et al. "Adversarial attacks and defenses on graphs." ACM SIGKDD Explorations Newsletter 22.2 (2021): 19-34.
> > > > >
> > > > > [19] Wang, Shoujin, et al. "Graph learning based recommender systems: A review." arXiv preprint arXiv:2105.06339 (2021).
> > > > >
> > > > > [20] Yasunaga, Michihiro, et al. "QA-GNN: Reasoning with language models and knowledge graphs for question answering." arXiv preprint arXiv:2104.06378 (2021).

---

> ### Author Response · Authors · 2022-11-18
> **Requesting feedback**
>
> Thank you once again for your review. We have tried to carefully address your concerns in our responses and it would be very valuable to us if you could provide your feedback. If any issues still remain that need to be resolved, please let us know.

---

> ### Author Response · Authors · 2022-12-02
> **Requesting feedback**
>
> Thank you once again for your review. We have tried to carefully address your concerns in our responses and it would be very valuable to us if you could provide your feedback. If any issues still remain that need to be resolved, please let us know.

---

### Decision · Program_Chairs · 2023-01-20

**Decision:**

Reject

**Justification For Why Not Higher Score:**

Based on the description above, there are various points the authors can improve.

**Justification For Why Not Lower Score:**

N/A

**Metareview: Summary, Strengths And Weaknesses:**

The authors study the robustness of graph learning techniques, specifically in the setting of time evolving graphs. For this, the paper proposes an adversarial attack approach (here: projected gradient descent under a specific budget constraint taking adjacent snapshots into account) showcasing the methods (non)robustness. The work has received mixed scores, with different opinions about its novelty, motivation in general, presentation, and experimental execution. Due to one strongly diverging score, the paper has been checked in detail again.

Regarding the motivation: Overall, I feel that the motivation is good and showcasing limitations of temporal methods is important.

Regarding the novelty: It has to be noted that attacks to temporal graphs have been studied before. Likewise have versions of PGD been studied for (graph) attacks multiple times. Thus, I feel that the technical novelty is indeed not the strongest. Moreover, at the end, the proposed constraint it still a usual budget constraint but just uses some "data-dependent" threshold.

Regarding experiments: Given that the constraint is still some budget constraint, I was wondering why the authors do not compare against a standard PGD attack (e.g. as shown in Table 1 first row). While not designed for dynamic data, one could still use PGD on every snapshot individually. At the moment, the only real competitor is TGA. The other baselines are, not surprisingly, not very strong.

Regarding presentation: I found the overall presentation to be fine. However, I don't really see the benefit of the presented theorems (partially because the statements are a bit obvious).

Given these points, I encourage the authors to revise the paper and resubmit at a future venue.